# Role of Hakai in m⁶A modification pathway in *Drosophila*

Yanhua Wang[1,2,5], Lifeng Zhang[1,2,5], Hang Ren[1,2,5], Lijuan Ma[1,2], Jian Guo[1], Decai Mao[3], Zhongwen Lu[1], Lijun Lu[1] & Dong Yan[4✉]

N6-methyladenosine (m⁶A), the most abundant internal modification in eukaryotic mRNA, is installed by a multi-component writer complex; however, the exact roles of each component remain poorly understood. Here we show that a potential E3 ubiquitin ligase Hakai colocalizes and interacts with other m⁶A writer components, and *Hakai* mutants exhibit typical m⁶A pathway defects in *Drosophila*, such as lowered m⁶A levels in mRNA, aberrant *Sxl* alternative splicing, wing and behavior defects. Hakai, Vir, Fl(2)d and Flacc form a stable complex, and disruption of either Hakai, Vir or Fl(2)d led to the degradation of the other three components. Furthermore, MeRIP-seq indicates that the effective m⁶A modification is mostly distributed in 5′ UTRs in *Drosophila*, in contrast to the mammalian system. Interestingly, we demonstrate that m⁶A modification is deposited onto the *Sxl* mRNA in a sex-specific fashion, which depends on the m⁶A writer. Together, our work not only advances the understanding of mechanism and regulation of the m⁶A writer complex, but also provides insights into how Sxl cooperate with the m⁶A pathway to control its own splicing.

[1] CAS Key Laboratory of Insect Developmental and Evolutionary Biology, CAS Center for Excellence in Molecular Plant Sciences, Institute of Plant Physiology and Ecology, Chinese Academy of Sciences, Shanghai, China. [2] CAS Center for Excellence in Biotic Interactions, University of Chinese Academy of Sciences, Beijing, China. [3] Gene Regulatory Lab, School of Medicine, Tsinghua University, Beijing, China. [4] State Key Laboratory of Genetic Engineering, School of Life Sciences, Fudan University, Shanghai, China. [5]These authors contributed equally: Yanhua Wang, Lifeng Zhang, Hang Ren. ✉email: yandong@fudan.edu.cn

There are a variety of chemical modifications on biological macromolecules, such as proteins, nucleic acids, and glycolipids. Like DNA methylation and histone modification, RNA modification represents an extra layer of epigenetic regulatory mechanism[1,2]. More than 150 chemical modifications in RNA have been discovered, and their biological functions are only starting to be revealed[3]. Chemical modifications of RNA exist in all organisms and for all forms of RNA, including tRNA, rRNA, mRNA, and long noncoding RNA. Common RNA modifications include N6-methyladenosine (m6A), N6,2'-O-dimethyladenosine (m6Am), N1-methyladenosine (m1A), 5-methylcytidine (m5C), N4-acetylcytidine (ac4C), 7-methylguanosine (m7G), and pseudouridine (Ψ), etc[4,5]. Among them, m6A is the most abundant internal modification of mRNA in eukaryotes. Although m6A in mRNA was found more than 40 years ago[6,7], it was only recently that the field has made extensive progress owing to technological and experimental breakthroughs. By combining m6A-specific antibody and high-throughput sequencing, MeRIP-Seq or m6A-Seq allows the m6A mapping at the whole transcriptome level, thereby providing the possibility to correlate RNA modifications with their biological functions[8,9]. These and subsequent studies revealed that m6A sites contain a consensus motif RRACH (R = G/A; H = U/A/C), and m6A peaks are enriched in the 3' untranslated region (UTR) and near the stop codon in yeast and mammals[8–10]. In Arabidopsis, m6A is enriched not only in 3'UTRs and near the stop codon but also in 5'UTRs and around the start codon[11]. In mammalian cells, m6A also accumulates in the 5'UTR region in response to stress conditions such as heat shock[12,13]. The distribution of m6A is important since it implies the mechanism by which m6A modification regulates its mRNA.

Another major breakthrough is the gradual elucidation of the m6A modification pathway by biochemical and genetic studies. The m6A is deposited by a multicomponent methyltransferase complex ("writers")[14–16], mainly recognized by YTH domain-containing "readers"[17], and can be removed by FTO and ALKBH5 "erasers"[18,19], although FTO was also indicated as an m6Am demethylase[20]. The key catalytic component of the m6A writer complex, Mettl3, was purified and cloned in 1990s[21,22]. Since then, studies from yeast, Arabidopsis, Drosophila, and mammalian cells have identified several core components of the writer complex[23,24], including Mettl14[25,26], WTAP (Fl(2)d)[27–29], VIRMA (Virilizer)[30,31], RBM15/15B (Spenito)[32,33], ZC3H13 (Flacc or Xio)[34–36], and Hakai[37]. Interestingly, Fl(2)d[38], Virilizer (Vir)[39], Spenito (Nito)[40], and Xio[36] were first identified from Drosophila sex determination screens and later realized as part of the writer complex. They regulate Drosophila sex determination by controlling the alternative splicing of the master regulatory gene Sex-lethal (Sxl)[41–46]. Recently, Mettl3, Mettl14, as well as the reader Ythdc1, were also shown to be involved in this process[33,47,48]. However, the detailed mechanism of how the m6A modification cooperates with Sxl protein to modulate its own splicing is still unclear. Thus, Drosophila can serve as a unique system to screen components in the m6A pathway and pinpoints a critical role for m6A in regulating splicing. Other than Sxl splicing, Drosophila m6A genes are highly expressed in the nervous system and exhibit similar wing and behavior defects when mutated[33,36,47,48]. Mutants of several fly m6A factors are viable and thus provide an ideal model to study other processes, such as metabolism and immunity, in the future.

Hakai, also known as CBLL1, was found as an interacting protein with several m6A writer components in proteomic studies[23,31,49]. It encodes a RING finger-type E3 ubiquitin ligase and was originally identified as an E-cadherin-binding protein in human cell lines[50]. It was proposed that Hakai ubiquitinates E-cadherin at the plasma membrane and induces its endocytosis, thus playing a negative role post-translationally. Due to the key role of E-cadherin in tumor metastasis, especially epithelial–mesenchymal transition, Hakai has been extensively studied mainly using cell culture and over-expression system[51], but a previous study using the Drosophila model did not observe an increase of E-cadherin level in Hakai mutants[52]. In Arabidopsis, Hakai mutants show partially reduced m6A levels and the mutant phenotypes are weaker than other writer components[37]. Importantly, the in vivo role of Hakai as a core m6A writer component has not been studied in any animal species. Here, we analyzed the role of Hakai in the Drosophila m6A modification pathway. Our results demonstrated that Hakai is a bona fide member of the m6A writer complex, with its mutants showing reduced global m6A levels, typical m6A mutant phenotypes, and commonly-regulated gene sets. We also obtained a high-quality fly m6A methylome using stringent MeRIP-seq, discovered a female-specific m6A methylation pattern for Sxl mRNA, characterized the role of Hakai in the m6A writer complex, and finally revisited the function of Hakai in E-cadherin regulation.

## Results

**Hakai interacts and colocalizes with known m6A writer complex subunits.** Since Hakai was found as an interacting protein with other m6A writer components in the mammalian proteomic study[23], we searched the large-scale Drosophila Protein interaction Map (DPiM) database[53]. Hakai as bait can pull down Fl(2)d, Vir, and Nito in affinity purification and mass spectrometry experiments; on the other hand, Flacc as a bait can pull down Hakai (Supplementary Fig. 1b). Similarly, our own previous mass-spec study using Fl(2)d or Nito as bait can reciprocally pull down Hakai (Supplementary Fig. 1a)[36]. To confirm these interactions, we performed both co-localization and co-immunoprecipitation (Co-IP) assays. GFP-Hakai localized to nucleus in live S2 cells and co-localized well with mRFP-Mettl3, mRFP-Mettl14, mRFP-Fl(2)d, mRFP-Nito, and mRFP-Flacc (Fig. 1a–e). Next, we transfected GFP-Hakai and different HA-tagged constructs in S2 cells, and used myc-GFP as a control. In the Co-IP experiments, GFP-Hakai, but not myc-GFP, was able to pull down HA-Mettl3, HA-Mettl14, HA-Nito, HA-Fl(2)d, and HA-Flacc (Fig. 1f). Interestingly, the pulldown between Hakai and Fl(2)d was particularly strong compared to other factors, suggesting that Hakai may directly interact with Fl(2)d, while the interaction between Hakai and Nito was the weakest (Fig. 1f). We then further examined the ability of GFP-Hakai to pull down endogenous Fl(2)d proteins using the available monoclonal antibody[42]. Hakai transcript is alternatively spliced, producing long and short protein isoforms (Fig. 2a), both of which contain a C3HC4 RING finger domain and a C2H2 zinc-finger domain (Supplementary Fig. 2a). We also included the short isoform in our assay and found that both GFP-Hakai (long isoform) and GFP-Hakai-S (short isoform) can robustly pull down Fl(2)d to a similar extent (Fig. 1g). Together, these data suggest that Hakai is a conserved core component of the m6A writer complex and its N-terminal domains are important for its interaction.

**Hakai is required to maintain proper levels of m6A methylation.** Hakai transcript shows a similar expression pattern to those of other m6A writers and readers[36], with high expression in the CNS, ovary, fat body and imaginal discs (Supplementary Fig. 2b; modENCODE developmental and tissue expression database[54]). During development, its expression is high in early embryos, decreases during larval stages, and rises again at pupal stages (Supplementary Fig. 2b), which coincides with the reported m6A levels[33]. To monitor endogenous protein expression, we raised an antibody against Hakai (Fig. 2a) and found that it is a ubiquitously-expressed nuclear protein that strongly colocalizes with Fl(2)d (Fig. 2c–c"). To further investigate Hakai function, we

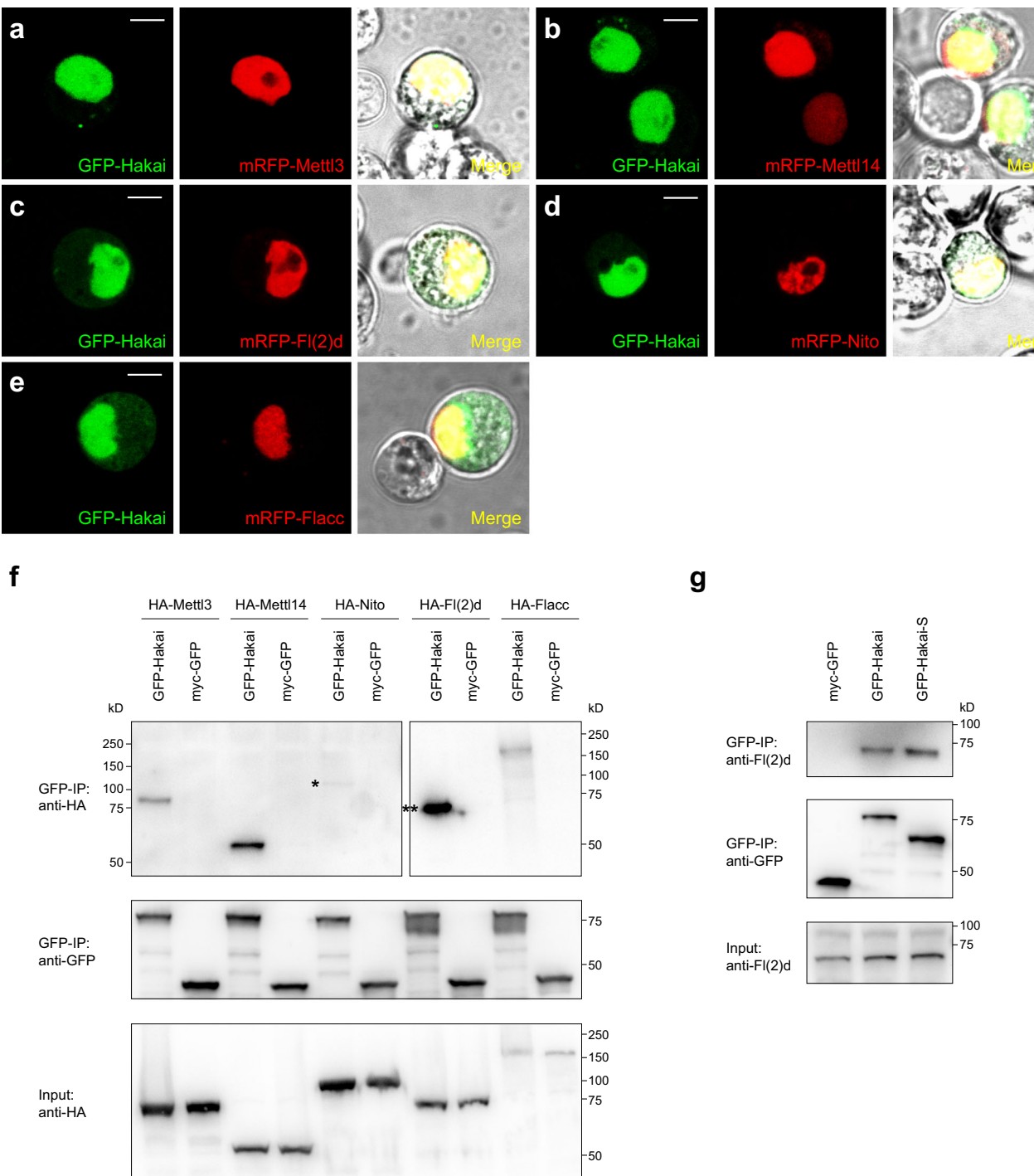

**Fig. 1 Hakai colocalizes and interacts with other m⁶A writer components. a–e** GFP-Hakai and mRFP-Mettl3, mRFP-Mettl14, mRFP-Fl(2)d, mRFP-Nito, or mRFP-Flacc were co-transfected into S2 cells and their subcellular localization examined in live conditions. All the proteins were predominantly nuclear and GFP-Hakai showed strong co-localization with other factors. Scale bars: 5 μm. **f** GFP-Hakai or myc-GFP and HA-Mettl3, HA-Mettl14, HA-Nito, HA-Fl(2)d, HA-Flacc were co-transfected into S2 cells. Cell lysates were immunoprecipitated using GFP nanobody and analyzed by western blot. myc-GFP was used as a control. GFP-Hakai can pull down HA-Mettl3, HA-Mettl14, HA-Nito, HA-Fl(2)d, and HA-Flacc. Note that much more HA-Fl(2)d was co-IPed than other factors (double asterisk), while the interaction with Nito was the weakest (asterisk). **g** myc-GFP, GFP-Hakai, or GFP-Hakai-S (short isoform) were transfected into S2 cells. Cell lysates were immunoprecipitated using GFP nanobody and analyzed by western blot. GFP-Hakai and GFP-Hakai-S can pull down endogenous Fl(2)d protein to a similar level. Source data are provided as a Source Data file. The experiments in **a–g** were repeated at least twice independently with similar results.

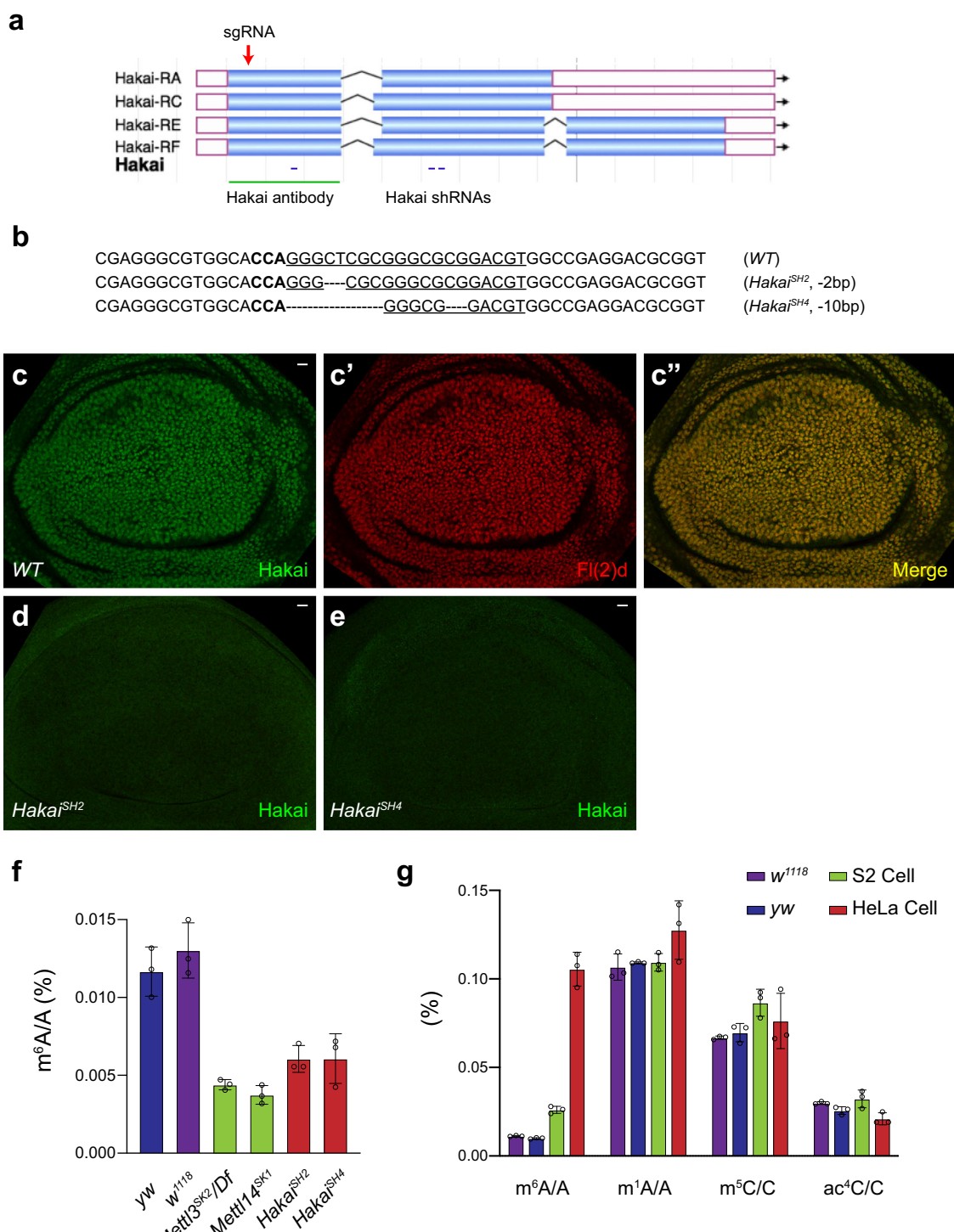

**Fig. 2 Hakai is required to maintain proper levels of m⁶A methylation. a** Flybase JBrowse view of the *Hakai* gene locus. *Hakai* has four transcripts due to alternative splicing that generates two long and two short protein isoforms. The positions of three independent shRNAs, *Hakai* sgRNA, and the protein region used to generate a Hakai antibody are indicated. **b** Sequencing results showing frameshift indels in *Hakai^SH2^* and *Hakai^SH4^* flies generated by CRISPR/Cas9-mediated mutagenesis. The targeted genomic DNA sequence is underlined and the NGG PAM sequence is in bold type. **c–c″** Hakai and Fl(2)d antibody staining in WT wing discs showing a high degree of co-localization. Hakai antibody staining was strongly reduced in *Hakai^SH2^* (**d**) or *Hakai^SH4^* (**e**) homozygous mutant wing discs. The experiments in **c–e** were repeated at least twice independently with similar results, and each time around 30 wing discs for any genotype were examined. Scale bars: 10 μm. **f** Quantifications of m⁶A relative to A in mRNA extracted from male adult flies by LC-MS. Compared to *yw* and *w^1118^* controls, m⁶A levels dropped to about 30% in *Mettl3* or *Mettl14* mutants and to <50% in *Hakai^SH2^* or *Hakai^SH4^* mutants. **g** Quantifications of m⁶A/A, m¹A/A, m⁵C/C, and ac⁴C/C levels in mRNA extracted from *yw* and *w^1118^* male flies, *Drosophila* S2 cells, and human HeLa cells. Note the substantial difference between fly and human m⁶A levels. **f, g** Data are presented as mean ± SD from three biological replicates. Source data are provided as a Source Data file.

generated a sgRNA and constructed or obtained three non-overlapping shRNA lines (Fig. 2a)[55,56]. By crossing the U6:3-Hakai sgRNA with nanos-Cas9[57], we generated a series of Hakai mutants with various small deletions and/or insertions. We chose two alleles, Hakai^SH2 and Hakai^SH4, for further analysis, since they represent different early frameshift mutations that are expected to disrupt translation (Fig. 2b). Indeed, only the background level of Hakai antibody staining remained in Hakai homozygous mutant wing discs compared to wild-type (Fig. 2d, e, compared to c), suggesting these are null or strong loss-of-function alleles. Hakai homozygous mutants are semi-lethal and only produce viable adult flies in noncrowded conditions, which have delayed developmental time, smaller size, and reduced lifespan.

We then measured N6-methyladenosine levels in Hakai mutant adults by quantitative liquid chromatography–mass spectrometry (LC-MS) and used yw and w^1118 as wild-type controls. We used an external calibration curve prepared with A and m^6A standards to determine the absolute quantities of each ribonucleoside (Supplementary Fig. 3). After two rounds of polyA selection, m^6A levels dropped to around 30% in Mettl3 or Mettl14 mutant flies, while m^6A levels reduced more than half in Hakai^SH2 or Hakai^SH4 mutants (Fig. 2f). These results clearly indicate a critical role of Hakai in m^6A methylation. During our analysis, we found that the m^6A levels we measured in Drosophila (0.01–0.02% of adenosine after two rounds of polyA purification) were one magnitude lower than those in mammals (0.1–0.4% of adenosine)[1]. This result is consistent with a previous study showing that m^6A accounts for 0.04% of adenosine after one round of polyA selection in yw flies[47]. Thus, we further measured the m^6A level, as well as the level of several other RNA modifications such as m^1A, m^5C, and ac^4C, in w^1118, yw flies, S2 cells, and human HeLa cells. To our surprise, m^6A level was five to ten times higher in human cells than those in Drosophila, while m^1A, m^5C, and ac^4C levels were comparable (Fig. 2g). These results imply that the function and mechanism of the m^6A pathway may be quite different between human and fly.

**Sxl alternative splicing and adult fly behavior were defective in Hakai mutant.** In Drosophila, Sxl pre-mRNA is the best-characterized example of m^6A-modified transcripts. Sxl transcripts are alternatively spliced. While the male form includes exon3 that contains a stop codon and leads to early termination of Sxl protein translation, the female form skips exon3 and thus produces a functional Sxl protein (Fig. 3a)[58]. Previously multiple m^6A sites have been mapped in introns on both sides of exon3 and these modifications were proposed to facilitate the alternative splicing of Sxl in female flies[47,48]. Indeed, the switch from female form to male form of Sxl splicing occurs in all m^6A mutants, including Mettl3, Mettl14, Ythdc1, Fl(2)d, Vir, Nito, Flacc, thus representing a gold standard to validate components in this pathway[33,34,36,38,41,43,47,48]. To monitor Sxl splicing pattern, we used a pair of primers flanking exon3 that detects the small female and large male spliced Sxl products in RT-PCR (Fig. 3a, b, lanes 1, 2)[59]. As positive controls, Sxl splicing was partially shifted from the female form to the male form in Mettl3 or Mettl14 mutant females (Fig. 3b, lanes 3–6, arrowheads). In Hakai^SH2 or Hakai^SH4 female flies, a large band corresponding to the male-specific spliced form was clearly detected (Fig. 3b, last 4 lanes, arrowheads), similar to Mettl3 or Mettl14 mutants.

Disruption of several m^6A components Fl(2)d, Vir, Nito or Flacc leads to not only aberrant Sxl splicing but also strongly reduced Sxl protein levels, thus generating a striking female-to-male transformation phenotype in adult flies[34,36,38,41,43,60].

Mutation of other factors Mettl3, Mettl14, or Ythdc1 alone does not affect Sxl protein levels and does not exhibit the transformation phenotype[33,47,48]. We found that Sxl protein level was not reduced in Hakai^SH2 or Hakai^SH4 female discs (Supplementary Fig. 4a–d). Furthermore, we expressed three Hakai RNAi using dome-Gal4 and did not observe any transformation phenotype in females, as evidenced before[36,41].

Other than Sxl, it was reported that splicing of several other genes, including Dsp1, CG8929, Aldh-III, and fl(2)d, depends on the m^6A pathway[33,34]. We then analyzed the splicing isoforms for these transcripts by RT-qPCR. In Hakai^SH4 mutants, the splicing patterns for all four genes were affected similarly to those in Mettl3 or Mettl14 mutants (Fig. 3c, see Supplementary Fig. 9f–i for positions of primers used). It is worth noting that in all four cases, the spliced isoforms were increased while the unspliced forms were reduced in m^6A pathway mutants.

Besides sex determination, m^6A writer and reader mutants show characteristic adult defects. The most prominent ones are the held-out wings and flightless phenotypes in Mettl3, Mettl14, Ythdc1, or Flacc mutants[36,47,48], likely due to loss of m^6A functions in the nervous system[33]. Wild-type flies normally keep their wings in a folded position (Fig. 3d, h), however, the majority of Mettl3 mutant flies cannot fold their wings correctly and exhibit held-out wings (Fig. 3e, h), and 100% Mettl3 mutant flies cannot fly (Fig. 3i). Interestingly, Hakai^SH2 and Hakai^SH4 mutants phenocopy Mettl3 adult defects in terms of the strong held-out wing (Fig. 3f–h) and 100% flightless phenotypes (Fig. 3i). Together, these data suggest that Hakai plays an important role in m^6A-modification pathway in vivo.

**Hakai, Vir, Fl(2)d, and Flacc form a stable complex.** We aim to investigate further the mechanisms of Hakai in m^6A methylation. Since Hakai is a potential E3 ubiquitin ligase, we examined the protein distribution of other m^6A writer factors in the absence of Hakai. For this purpose, we generated antibodies against Mettl3, Mettl14, and Vir, constituting a full toolkit for all seven Drosophila m^6A writers. In wild-type wing discs, all seven m^6A writer components appeared as ubiquitous nuclear proteins that colocalizes extensively with each other (Figs. 4a–c, 2c–c" and Supplementary Fig. 5a, c). Expression of Hakai RNAi in the dorsal half of the wing disc using ap-Gal4 led to no effect on Mettl3, Mettl14, and Nito protein levels (Fig. 4j–l), but the strong reduction of Fl(2)d, Vir, and Flacc levels (Fig. 4h–i). In addition, we crossed actin-Cas9 with U6-Hakai-sgRNA flies to generate random Hakai loss-of-function clones. These clones were marked by the loss of Hakai staining, and the Fl(2)d level was reduced in these clones as well (Fig. 4f–f'). Since the roles of several other m^6A writer components are not fully understood, we extended our immunostaining assays to those genes. Interestingly, knocking down vir by RNAi resulted in no effect on Mettl3, Mettl14, and Nito protein levels (Fig. 4q, n', r), but strong reduction of Fl(2)d, Hakai, and Flacc levels (Fig. 4n, o, p). Similarly, depletion of fl(2)d by RNAi did not change the protein levels of Mettl3, Mettl14, and Nito (Fig. 4s', w, x), but strongly reduced Hakai, Vir, and Flacc levels (Fig. 4t–v). Together, these results suggest that Fl(2)d, Vir, Hakai, and Flacc form a stable complex, and disruption of either Fl(2)d, Vir, or Hakai leads to degradation of the whole complex, while Mettl3, Mettl14, and Nito are not part of this complex. In cell culture, ZC3H13 plays a role in anchoring the writer complex in the nucleus[35]. Consistent with this, we observed more diffusive and less nuclear staining of Fl(2)d upon flacc RNAi knockdown in the wing discs (Fig. 4d, e, compare insets in 4e with 4h). Based on our data, we proposed a working model for the m^6A writer complex (Fig. 4y, see "Discussion").

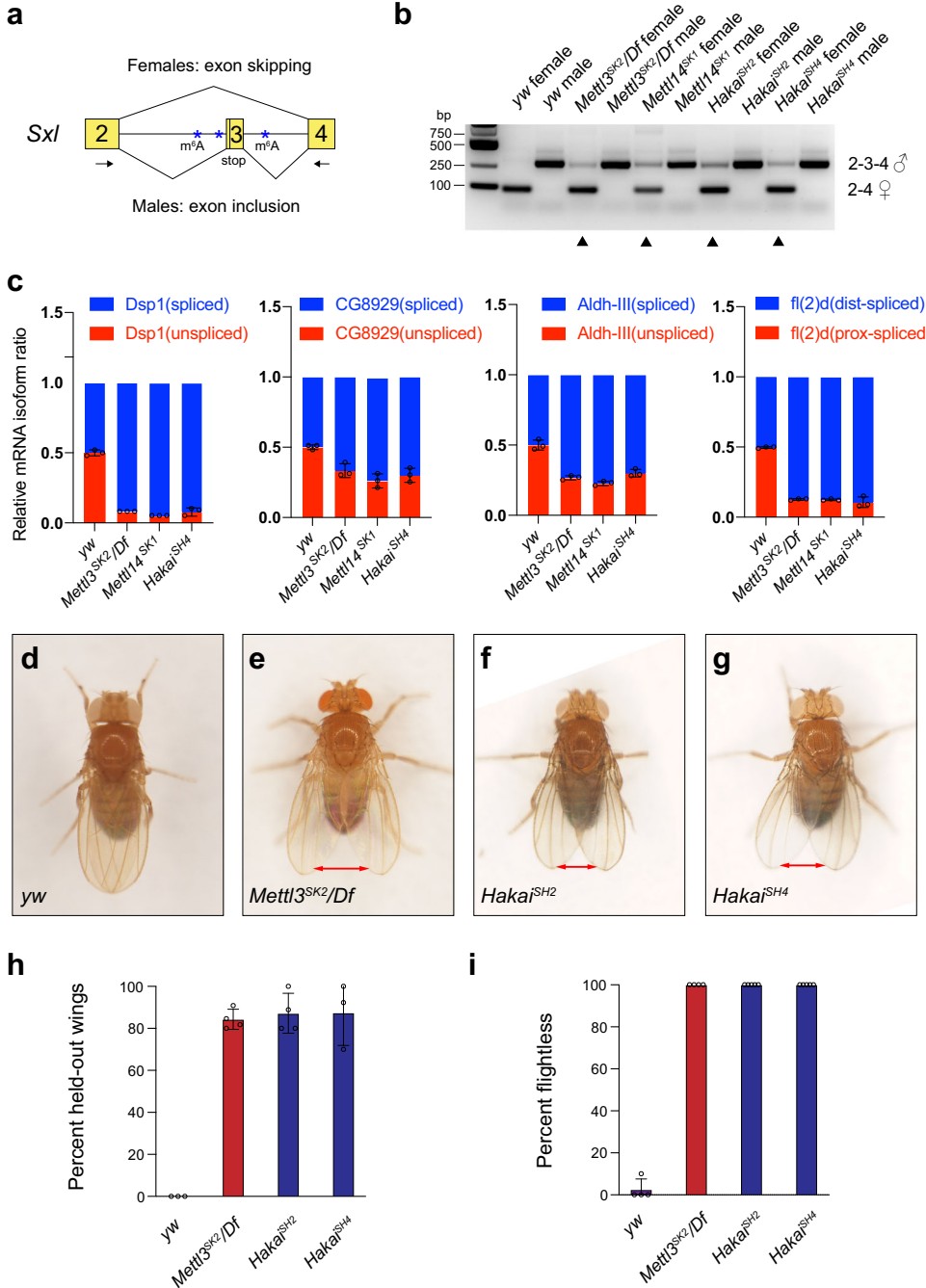

**Fig. 3 Hakai regulates *Sxl* alternative splicing and adult fly behavior. a** Diagram showing the alternative splicing event that produces male- or female-specific *Sxl* transcripts. The arrows indicate primers used for RT-PCR. **b** *Sxl* splicing was analyzed by RT-PCR using RNAs extracted from adult flies of indicated genotypes. Note the appearance of male-specific bands in *Mettl3SK2/Df*, *Mettl14SK1*, *HakaiSH2*, *HakaiSH4* females (arrowheads). Male-specific bands: 2–3–4. Female-specific bands: 2–4. **c** Relative isoform quantification for *Dsp1*, *CG8929*, *Aldh-III* and *fl(2)d* by qPCR in control or m6A mutant flies. Hakai is involved in m6A-dependent splicing events. Data are presented as mean ± SD from three biological replicates. **d** *yw* flies have their wings properly folded. (**e**) *Mettl3SK2/Df*, (**f**) *HakaiSH2*, (**g**) *HakaiSH4* flies cannot fold their wings and exhibit a held-out wing phenotype (marked by the double arrows). The frequency of flies showing held-out wings were quantified in (**h**). **i** Flies of the indicated genotypes were tested for their flight abilities, and the number of flightless flies was quantified. All flies used from (**c**) to (**i**) were males. **h–i** Data are presented as mean ± SD from three to five biologically independent groups (**h**, *n* = 3, 4, 4, 3; **i**, *n* = 4, 4, 5, 5). Source data are provided as a Source Data file.

**Hakai does not mediate E-cadherin levels in wing discs.** Hakai was first demonstrated as an E-cadherin interaction protein and its role in E-cadherin endocytosis and down-regulation was extensively studied in cell cultures[50,51]. However, a previous study in *Drosophila* failed to observe a major role for Hakai in E-cadherin regulation[52]. Thus, we addressed this question using our genetic toolset. In wild-type wing-disc epithelia, E-cadherin

showed a membrane distribution and accumulates in the adherens junction (Fig. 5a, a'). First, we used *ap-Gal4* to drive the expression of *Hakai* RNAi in the dorsal half of the wing disc. Although Hakai level was effectively knocked down (Fig. 5c'), E-cadherin level did not change in either the apical or the lateral section (Fig. 5c, c"). Second, in *Hakai* mutant clones generated by crossing *actin-Cas9* with *U6-Hakai-sgRNA*, E-cadherin level was

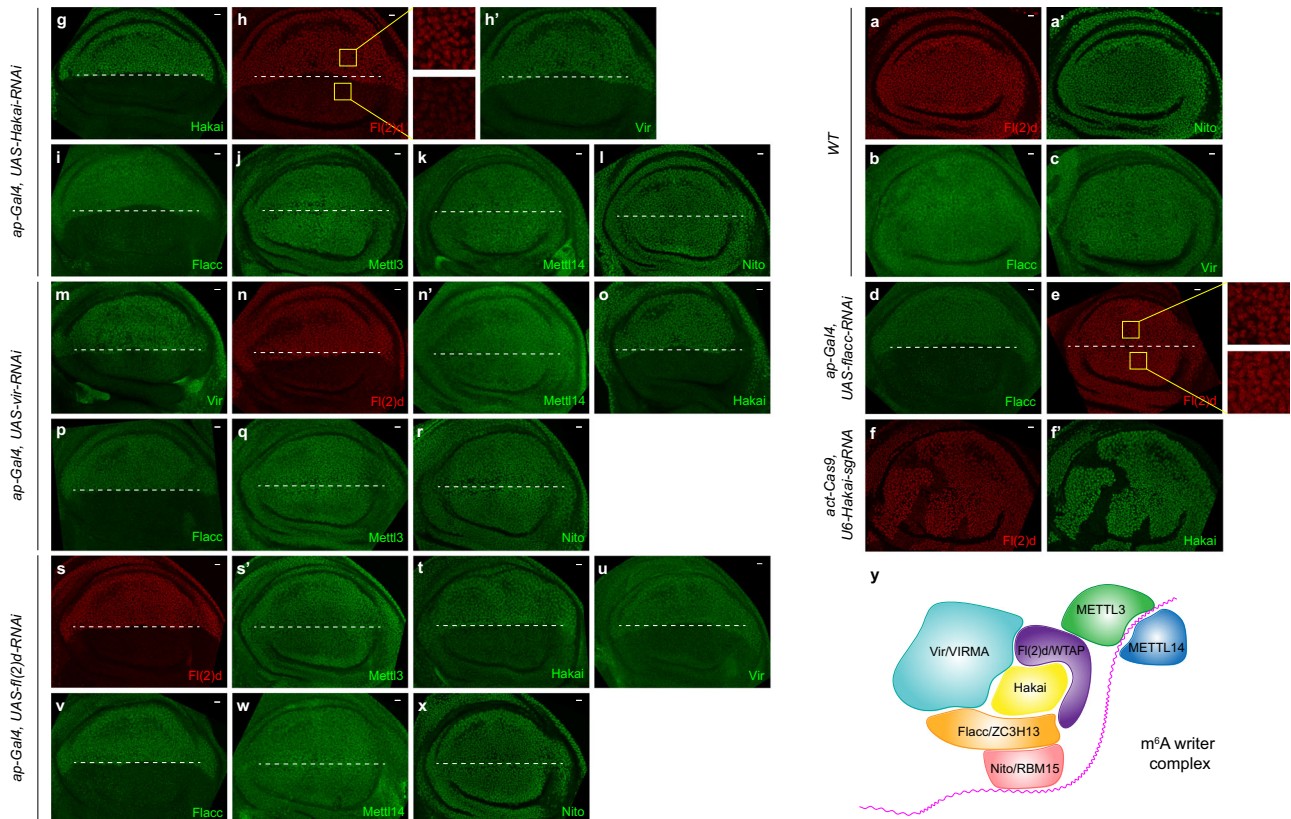

**Fig. 4 Hakai, Vir, Fl(2)d, and Flacc form a stable complex.** In wild-type wing discs, Fl(2)d (**a**), Nito (**a'**), Flacc (**b**), and Vir (**c**) show ubiquitous nuclear staining patterns. **g–l** Expressing *Hakai* RNAi in the dorsal half of the disc (below the dashed line) using *ap-Gal4* resulted in a strong reduction of Fl(2)d (**h**, squared areas are magnified on the right), Vir (**h'**), and Flacc (**i**) levels, but not Mettl3 (**j**), Mettl14 (**k**), and Nito staining (**l**). **f–f'** Fl(2)d staining is similarly reduced in *Hakai* mutant clones generated by *actin-Cas9/U6-Hakai-sgRNA* (**f**), which are marked by the loss of Hakai staining (**f'**). **m–r** Expressing *vir* RNAi in the dorsal half of the disc (below the dashed line) using *ap-Gal4* resulted in a strong reduction of Fl(2)d (**n**), Hakai (**o**), and Flacc (**p**) levels, but not Mettl3 (**q**), Mettl14 (**n'**), and Nito staining (**r**). **s–x** Expressing *fl(2)d* RNAi in the dorsal half of the disc (below the dashed line) using *ap-Gal4* resulted in a strong reduction of Hakai (**t**), Vir (**u**), and Flacc (**v**) levels, but not Mettl3 (**s'**), Mettl14 (**w**), and Nito staining (**x**). **d**, **e** Expressing *Flacc* RNAi in the dorsal half of the disc (below the dashed line) using *ap-Gal4* led to more diffusive and less nuclear staining of Fl(2)d (**e**, squared areas are magnified on the right). The experiments in **a**–**x** were repeated at least twice independently with similar results, and each time around 30 wing discs for any genotype were examined. Scale bars: 10 μm. **y** A working model of the m⁶A writer complex comprised of seven core components.

not affected either (Fig. 5d–d"). Finally, we examined the localization of E-cadherin and Hakai in detail using large tracheal cells. As shown in Fig. 5b–b", most E-cadherin staining was on the cell membrane, while most Hakai staining was in the nucleus, and we did not observe co-localization between these two proteins. In conclusion, Hakai is not important for E-cadherin levels in *Drosophila*, and its major function likely happens in the nucleus.

**The m⁶A modifications are deposited onto the *Sxl* mRNA in a sex-specific fashion.** Previously, only two studies were reported to map global m⁶A methylation pattern in *Drosophila*, one being MeRIP-seq in S2R + cells and the other being miCLIP in embryos[33,47]. To obtain a high-stringent m⁶A methylome in adult flies, we performed methylated RNA immunoprecipitation sequencing (MeRIP-seq) in *yw* male and female flies. We included two replicates for each MeRIP-seq and the mapped reads were enriched around TSS and TES in IP samples compared to inputs (Supplementary Fig. 6). A peak detection algorithm was used to identify m⁶A peaks (*P* < 0.05, Supplementary Data 1), which were enriched in the 3′UTR and close to the stop codon, and to a lesser extend enriched in the 5′UTR and around the start codon (Fig. 6a, b). De novo motif analysis using HOMER identified the consensus sequence RRACH (Fig. 6c), consistent with those in the mammalian system[8,9].

We then zoomed in on the *Sxl* locus, especially around the male-specific exon3, for potential m⁶A sites (Fig. 6d). Although the mRNAs used for MeRIP-seq were through polyA selection, we were able to detect 3–4 m⁶A peaks in and around exon3 (Fig. 6e). Strikingly, these m⁶A peaks only exist in female flies, but not males (Fig. 6e, compare ywF_IP with ywM_IP). We next validated the MeRIP-seq results with independent m⁶A-immunoprecipitation (IP)-qPCR. Four regions on *Sxl* mRNA, intron2–1, intron2–2, exon3, and intron3, were measured by qRT-PCR (Fig. 6e), and substantial enrichment was observed only in female mRNA IPed with m⁶A-specific antibody, but not in female mRNA IPed with control IgG, nor in male mRNA IPed with either m⁶A or IgG antibody (Fig. 6f). These results demonstrated that the m⁶A modifications are deposited in a sex-specific manner, which has not been shown in *Drosophila* or any other species before. We further investigated whether m⁶A modifications around *Sxl* exon3 are dependent on the m⁶A writer. In *Mettl3* mutant females, the enrichments on intron2-2, exon3, and intron3, but not intron2-1, were significantly reduced, probably implying the importance of the former three m⁶A sites (Fig. 6g).

How can these m⁶A modifications be installed only in females? In fact, Sxl protein itself is the master sex determination factor that is only expressed in females. Sxl binds to polyU sites located in *Sxl* intron2 and intron3[61,62] and interestingly our mapped m⁶A peaks were close to those Sxl-binding sites (Fig. 6e). In addition, it

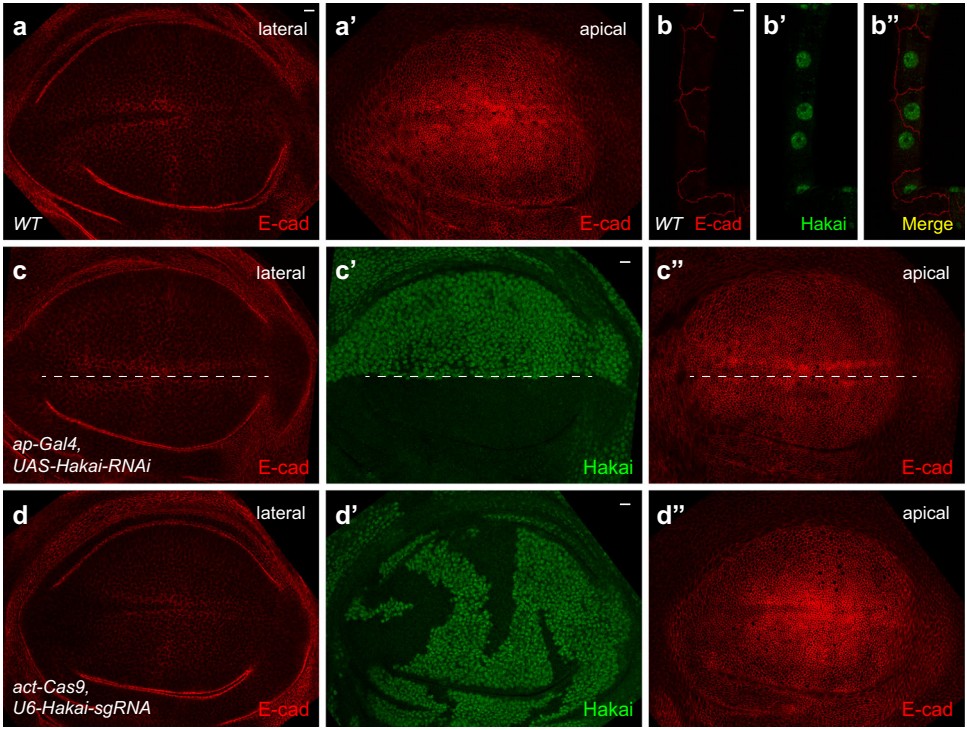

**Fig. 5 Hakai does not mediate E-cadherin levels in wing discs. a–a′** Lateral (**a**) and apical (**a′**) section of E-cadherin staining in wild-type wing-disc epithelia. **b–b″** E-cadherin is mainly localized in the cell membrane (**b**), while Hakai is mostly in the nucleus (**b′**) in the large tracheal cells. **c–c″** Expression of *Hakai* RNAi in the dorsal half of the disc (below the dashed line) using *ap-Gal4* led to a strong reduction of Hakai (**c′**), but E-cadherin staining was not affected either laterally (**c**) or apically (**c″**). **d–d″** Lateral (**d**) and apical (**d″**) E-cadherin distribution is not changed in *Hakai* mutant clones marked by the absence of Hakai staining (**d′**), when crossing *actin-Cas9* with *U6-Hakai-sgRNA*. The experiments in **a–d″** were repeated at least twice independently with similar results, and each time around 30 wing discs for any genotype were examined. Scale bars: 10 µm.

was known that Sxl physically interacts with four m⁶A writer components, Fl(2)d, Vir, Nito, and Flacc[36,41,42,45,46]. Therefore, based on our data, we developed a model to explain how the m⁶A modifications cooperate with Sxl protein to regulate its mRNA splicing (Fig. 6h). Sxl in females recruits the m⁶A writer complex that in turn methylates m⁶A sites located in exon3 and nearby introns. Since these sites are quite close to exon/intron junction regions, m⁶A readers may bind to these sites and interfere with the splicing machinery, forcing the exon3 to be skipped in females (Fig. 6h).

**Effective m⁶A modification occurs in 5′UTR and around start codon in *Drosophila*.** Since previous m⁶A-Seq data in *Drosophila* were generated only in wild-type cells or tissues[33,47], we extended our MeRIP-seq to *Mettl3*, *Mettl14*, and *Hakai* mutants in order to find out the effective m⁶A modifications that depend on the m⁶A writer (Supplementary Fig. 6 and Supplementary Data 1). We used previously reported alleles *Mettl3^{SK2}* and *Mettl14^{SK1}* for our experiments[47], since they are strong or null alleles as evidenced by the complete loss of Mettl3 and Mettl14 antibody staining in these mutant discs, respectively (Supplementary Fig. 5). Peak density plots indicated that m⁶A peaks were primarily enriched in the 3′UTR and near the stop codon, and enrichment in the 5′UTR and around the start codon seems to be reduced in these mutants compared to *yw* (Fig. 7a–d, compare to 6a). The cumulative distribution plot demonstrated that there is less fold enrichment (m⁶A-IP/input) in m⁶A mutants than *yw*, implying less methylation levels (Fig. 7e). Indeed, when we filtered m⁶A peaks with higher stringency (fold enrichment ≥5), the number of m⁶A peaks dropped >40% in *Mettl3* mutant compared with *yw* (Fig. 7f).

More importantly, we focused on those peaks changed upon m⁶A writer depletion. Interestingly, many more m⁶A peaks were reduced (*Mettl3*, 2745; *Mettl14*, 2615; *Hakai*, 2036; *P* < 0.05 and fold change ≤0.5) than increased (*Mettl3*, 330; *Mettl14*, 260; *Hakai*, 278, *P* < 0.05 and fold change ≥2) in these mutants, validating their role as m⁶A writers (Fig. 7g, h and Supplementary Data 2). Only in reduced m⁶A peaks, a significant overlap between the three mutants was found (1345 common peaks), suggesting that Hakai plays a similar role to Mettl3 and Mettl14 in m⁶A methylation. The common reduced peaks in three writer mutants likely represent high-confidence m⁶A modification sites in *Drosophila*. We focused on common reduced m⁶A peaks for further analysis. Surprisingly, around 90% of these peaks were located in the 5′UTR and close to the start codon, and only a small portion of them occurred in the 3′UTR (Fig. 7i). A few examples of loss of m⁶A peaks in the three mutants were shown in Fig. 7j and Supplementary Fig. 7, which almost always happened in the 5′UTR, while m⁶A peaks in the 3′UTR were generally not changed. We then used m⁶A-IP-qPCR to validate the reduction of m⁶A peaks in the 5′UTR region for 11 genes and found a significant reduction of m⁶A signal in *Mettl3* mutant versus *yw* (Fig. 7k). These results indicate that although most m⁶A sites map to 3′UTRs in *Drosophila*, the peaks responding to the loss of m⁶A writers, and thus the effective ones, are distributed in 5′UTRs and this is different from the mammalian system. The majority of the peaks in 3′UTRs may be mediated by another methyltransferase or come from a non-specific background.

Next, we analyzed gene expression regulated by m⁶A writer components by RNA-seq (Supplementary Data 3). In *Mettl3*, *Mettl14*, and *Hakai* mutant adult flies, 987, 954, and 886 genes were differentially expressed (*P* < 0.05 and fold change ≥2 or

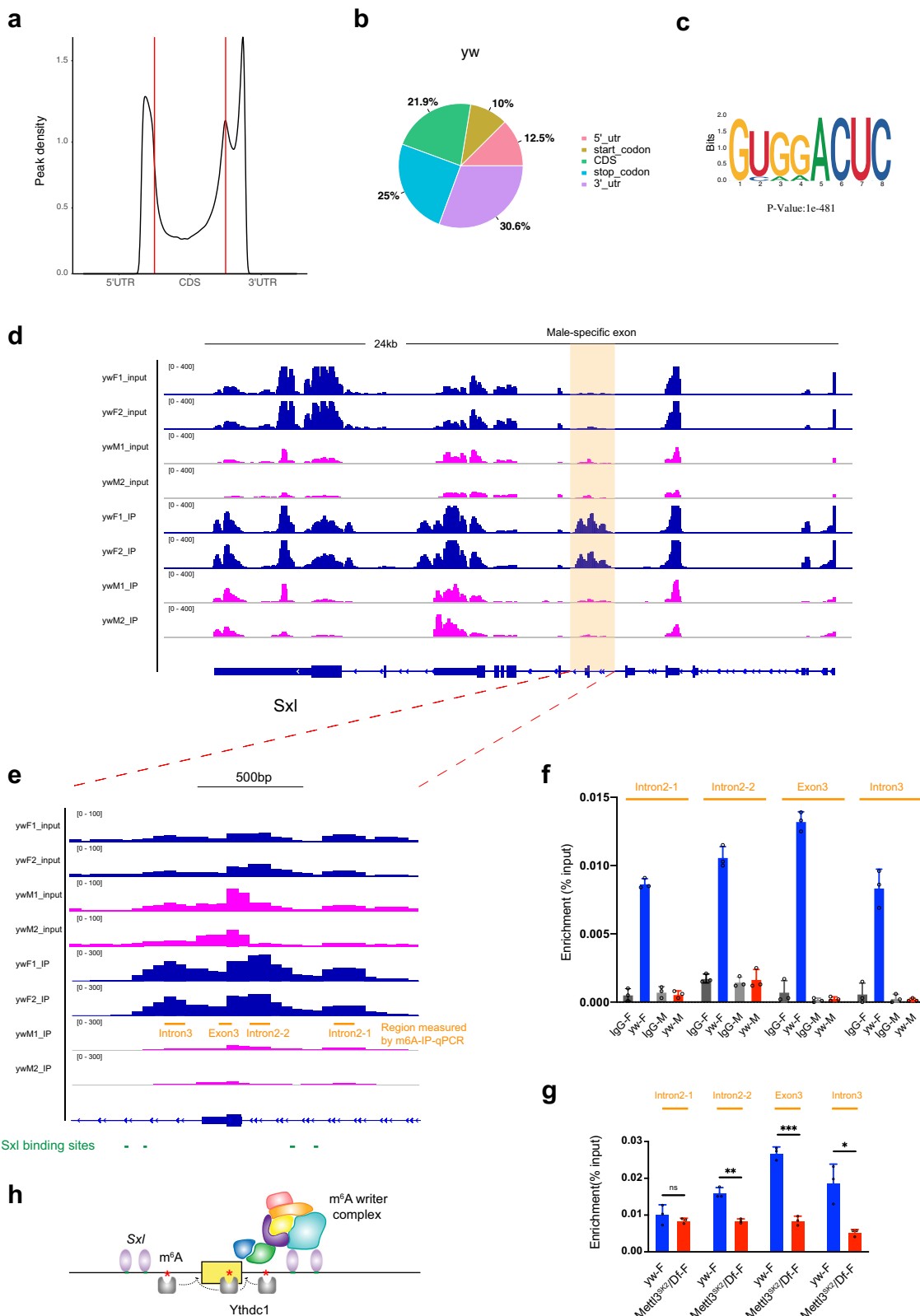

≤0.5), respectively (Supplementary Fig. 8a and Supplementary Data 4). These genes substantially overlapped with each other (Supplementary Fig. 8b) and common differentially expressed genes generally changed in the same pattern in the three mutants (Supplementary Fig. 8c), arguing that they act in the same pathway. Interestingly, differentially expressed genes were strongly enriched for immune response genes in the GO analysis

and for metabolic pathways in the KEGG analysis (Supplementary Fig. 8d, e). By exhibiting MeRIP-seq together with RNA-seq data, we observed the reduction of m6A peaks in *Mettl3*, *Mettl14*, or *Hakai* mutant, and no obvious trend for change of mRNA expression correlated with differential m6A peaks (Supplementary Fig. 8f–h and Supplementary Data 5). We further divided the genes into m6A targets or nontargets, and the cumulative plot

**Fig. 6 Female-specific deposition of m6A on the *Sxl* mRNA. a** MeRIP-seq shows that the normalized density of m6A peaks across 5′UTR, CDS, and 3′UTR of mRNA in *yw* male adult flies. **b** Pie charts depicting m6A peak distribution in different transcript segments in *yw* flies. **c** Sequence motif identified from m6A peaks in *yw* flies by HOMER program. **d** Integrative Genomics Viewer (IGV) tracks displaying MeRIP-seq (lower panels, IP) and RNA-seq (upper panels, input) reads along *Sxl* locus in *yw* male (ywM) and female flies (ywF). Two replicates are shown. The region around male-specific exon3 is shaded and enlarged in (**e**) highlighting 3–4 m6A peaks on and around exon3 only in female flies. Adjacent Sxl-binding sites are also shown. **f** m6A-IP-qPCR showing enrichments over *Sxl* mRNA in *yw* male or female flies IPed with m6A or IgG antibody. Regions measured are indicated in (**e**). **g** m6A-IP-qPCR showing enrichments over *Sxl* mRNA in *yw* or *Mettl3^SK2^/Df* female flies IPed with m6A antibody. Data are presented as mean ± SD from three biological replicates. *$P < 0.05$; **$P < 0.01$; ***$P < 0.001$; ****$P < 0.0001$; ns not significant, two-sided unpaired *t* test. In **g**, $P = 0.3679, 0.0012, 0.0001, 0.0114$. Source data are provided as a Source Data file. **h** A working model on how the m6A modifications cooperate with Sxl to regulate its mRNA splicing.

shows that there was no obvious difference between m6A targets and nontargets in *Hakai* or *Mettl3* mutant flies (Supplementary Fig. 8i, j), while there was a slight positive effect of m6A on mRNA levels in *Mettl14* mutant (Supplementary Fig. 8k), consistent with a previous report[33]. Moreover, we performed RNA decay assay for validated m6A-containing transcripts. Four hours after transcription inhibition by actinomycin D, we did not observe the significant difference in mRNA levels for these genes between *Mettl3* and *yw* imaginal discs (Fig. 7l). These findings are consistent with the notion that effective m6A modifications are located in 5′UTRs in *Drosophila*, and thus do not mediate mRNA degradation as in the mammalian system.

Finally, we performed a similar analysis for splicing changes. In *Mettl3*, *Mettl14*, and *Hakai* mutant flies, 445, 364, and 340 genes were differentially spliced (FDR < 0.05, IncLevelDifference ≥0.2 or ≤ −0.2), respectively, and they overlapped substantially with each other (Supplementary Fig. 9a, c and Supplementary Data 6–8). The differential alternative spliced events occurred in all splicing categories (Supplementary Fig. 9b) and were functionally enriched for mRNA splicing and signaling transduction, etc (Supplementary Fig. 9d, e). We then examined the alternatively spliced transcripts of *Dsp1*, *CG8929*, *Aldh-III*, and *fl(2)d*, which were regulated by the m6A pathway (Fig. 3c). In all four cases, we found heavily methylated peaks surrounding the splicing junctions that depended on Mettl3, suggesting that these modifications may regulate splicing directly (Supplementary Fig. 9f–i). Interestingly, it seemed that the role of m6A modification is to repress splicing in these events. Even in the case of *Sxl*, the m6A modification plays a role to inhibit splicing as well, whether this is a general mechanism needs to be determined in the future.

## Discussion

m6A modification has been known for more than 40 years[6] but has recently gained great attention due to the emergence of technologies to map m6A methylome[8,9,63], as well as the identification of the writers, readers, and erasers in this pathway[4,14–16]. Since the initial purification of the key methyltransferase Mettl3[21], other components of the writer complex were gradually identified through biochemical experiments and genetic screens. We now know that m6A writer complex is comprised of multiple components including Mettl3, Mettl14, WTAP, VIRMA, RBM15/15B, ZC3H13. Hakai was first indicated as a WTAP interaction protein[23] and was shown later to be required for full m6A methylation in *Arabidopsis*[37]; however, its role in the m6A pathway in animals has not been studied. Here, we show that Hakai interacts with other m6A writer subunits, and *Hakai* mutants exhibit characteristic m6A pathway phenotypes, such as lowered m6A levels in mRNA, aberrant alternative splicing of *Sxl* and other genes, held-out wings, and flightless flies, as well as reduced m6A peaks shared with *Mettl3* and *Mettl14* mutants in MeRIP-seq. Altogether, these data unambiguously argue that Hakai is the seventh, and likely last core component of the conserved m6A writer complex.

Each component in the m6A writer complex plays a role in mRNA methylation but their exact roles are not well understood[64]. Our systematic analysis of several m6A writer subunits has provided insights into the mechanism of this important complex. We found that Fl(2)d, Vir, Hakai, and Flacc form a stable complex, and knocking down either of Fl(2)d, Vir, or Hakai led to the degradation of the other three components. Mettl3, Mettl14, and Nito were not affected by the disruption of Fl(2)d, Vir or Hakai, suggesting that they have separate functions. Knocking down Flacc resulted in less nuclear staining of Fl(2)d, consistent with a role in nuclear localization of the writer complex[35]. Based on these results, we proposed a model for the m6A methyltransferase complex (Fig. 4y). Mettl3 and Mettl14 form a stable heterodimer to catalyze the addition of the methyl group to mRNA. Nito/RBM15 contains three RRM domains and binds to positions adjacent to m6A sites, thus may provide target specificity for the m6A writer complex[32]. Fl(2)–Vir–Hakai–Flacc form a platform to connect different components and may integrate environmental and cellular signals to regulate m6A methylation.

Hakai is a potential E3 ubiquitin ligase with an intact C3HC4 RING domain and a C2H2 domain. Its absence led to the degradation, rather than the accumulation of other m6A writer subunits, indicating that it may not act as an E3 ubiquitin ligase in this complex. Hakai was initially identified as an E-cadherin binding protein to downgrade its levels[50], and the role of Hakai in cell proliferation and tumor progression was extensively studied in cell culture[51]. However, our in vivo analysis using various genetic tools did not find a role of Hakai in E-cadherin regulation. In addition, Hakai appeared as a ubiquitous nuclear protein showing little co-localization with E-cadherin in the membrane. Consistently, Hakai was shown to interact with PTB-associated splicing factor (PSF), a nuclear protein, and to affect its RNA-binding ability[65]. Thus, the role of Hakai in E-cadherin regulation needs to be further investigated using the knockout mouse model and whether Hakai has other substrates for its E3 ligase activity also needs to be determined.

Recent emerging studies suggest that m6A is involved in numerous developmental processes and human diseases[16,66], mainly by regulating mRNA stability, translation, or splicing. Pioneer work from three labs has established the framework for the m6A pathway in *Drosophila*[33,47,48]. However, only published *Drosophila* m6A methylome was performed in S2R + cells or embryos[33,47] and was not done against writer mutants. Other than *Sxl*, few m6A target loci have been firmly mapped. By performing MeRIP-seq in wild-type adult flies as well as *Mettl3*, *Mettl14*, and *Hakai* mutants, we demonstrated that although most m6A peaks are distributed in 3′UTRs, the functional peaks responding to the loss of m6A writers are mainly located in 5′UTRs. This finding indicates a major difference between *Drosophila* and mammalian m6A methylome, which mainly occurs in 3′UTRs, and is in agreement with a recently published manuscript using miCLIP[67]. Interestingly, our LC-MS data show that the overall level of m6A modification in *Drosophila* only

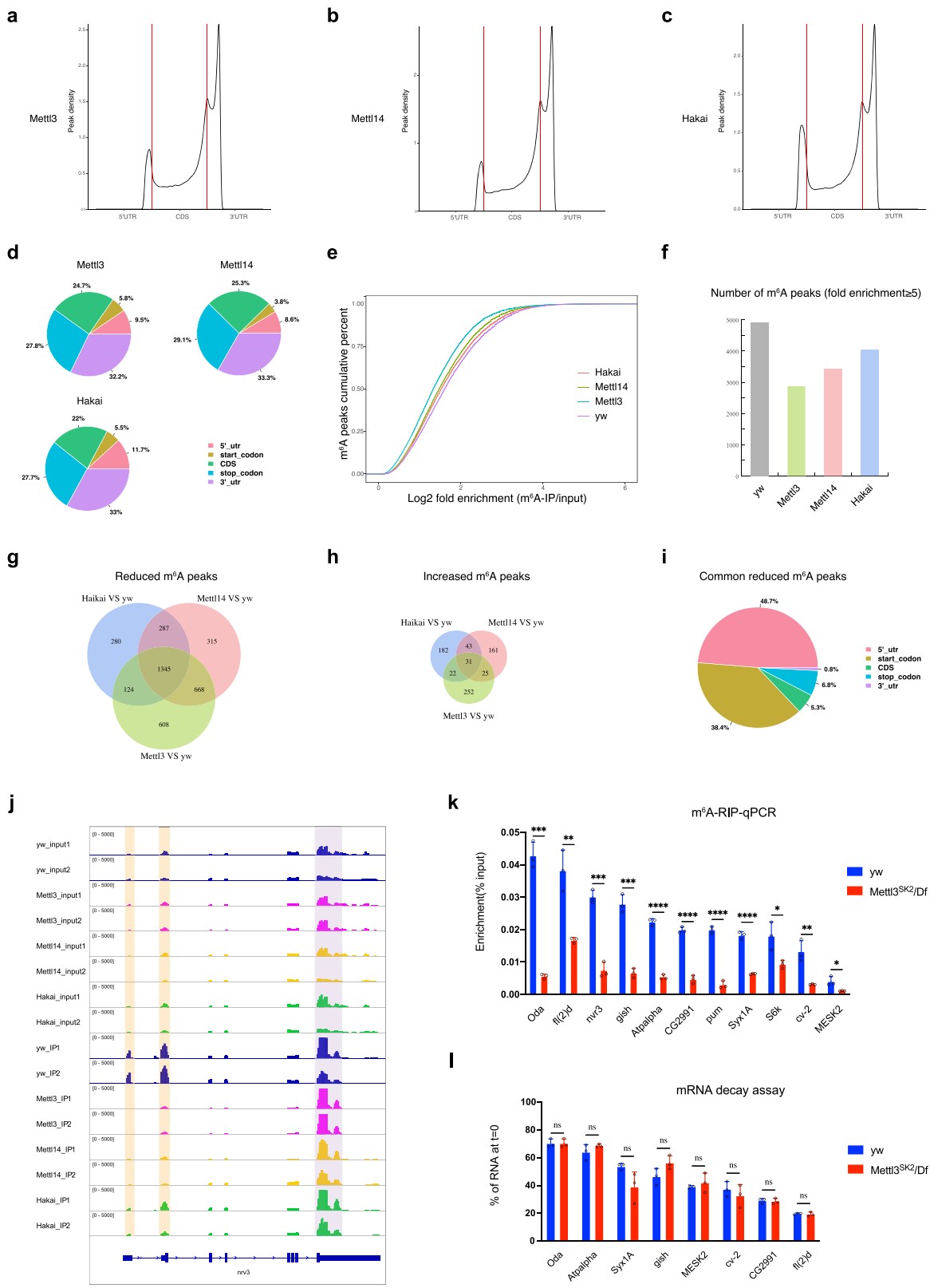

accounted for 10–20% of that in mammalian cells. *Mettl3* or *Mettl14* mutants are embryonic lethal in mice while they develop into adults in flies. It is possible that the m$^6$A pathway acquires additional functions during evolution.

m$^6$A modification in 3′UTRs usually causes mRNA instability and m$^6$A in 5′UTRs is linked to translation enhancement[1,4]. In agreement with the view that functional m$^6$A peaks are located in

5′UTRs in *Drosophila*, we did not observe an increase in mRNA half-life of m$^6$A targets in *Mettl3* mutants compared to wild-type. These results imply that the major role of m$^6$A modification in *Drosophila* is not on mRNA degradation, but possibly on translation upregulation, which can be tested by combining ribosome profiling[68] and functional analysis of a single transcript in the future. Our data by combining MeRIP-seq and splicing analysis

**Fig. 7 Effective m6A modification in *Drosophila* is distributed in 5′UTRs. a–c** MeRIP-seq shows the normalized density of m6A peaks across 5′UTR, CDS, and 3′UTR of mRNA in *Mettl3*, *Mettl14*, and *Hakai* mutant male adult flies. **d** Pie charts depicting m6A peak distribution in different transcript segments in *Mettl3*, *Mettl14*, and *Hakai* flies. **e** Cumulative percent distribution of log2 fold enrichment (m6A-IP/input) in *yw*, *Mettl3*, *Mettl14*, and *Hakai* flies. **f** When filtered with fold enrichment ≥5, the number of m6A peaks dropped in *Mettl3*, *Mettl14*, or *Hakai* mutant compared with *yw*. **g** Venn diagram showing the overlap of reduced m6A peaks (*P* < 0.05 and fold change ≤0.5) between *Mettl3*, *Mettl14*, and *Hakai* flies versus *yw* control. **h** Venn diagram showing the overlap of increased m6A peaks (*P* < 0.05 and fold change ≥2) between *Mettl3*, *Mettl14*, and *Hakai* flies versus *yw* control. **i** Pie chart depicting the distribution of 1345 common reduced m6A peaks in different transcript segments. **j** IGV tracks displaying MeRIP-seq (lower panels, IP) and RNA-seq (upper panels, input) reads along *nrv3* mRNA in *yw*, *Mettl3*, *Mettl14*, and *Hakai* male flies. Note the reduced peaks in the 5′UTR (shaded in yellow) and peaks in the 3′UTR (shaded in purple) are not changed. **k** m6A-IP-qPCR validation for selected mRNAs in *yw* or *Mettl3^{SK2}/Df* male flies. **l** The m6A-containing mRNAs were measured by qPCR in *yw* or *Mettl3^{SK2}/Df* imaginal discs, at time = 0 or after four hours actinomycin D treatment to block transcription. **k, l** Data are presented as mean ± SD from three biological replicates. *\*P* < 0.05; *\*\*P* < 0.01; *\*\*\*P* < 0.001; *\*\*\*\*P* < 0.0001; ns not significant, two-sided unpaired *t* test. In **k**, *P* = 0.0001, 0.0043, 0.0003, 0.0003, <0.0001, <0.0001, <0.0001, <0.0001, =0.0273, 0.0073, 0.0276 for *Oda*, *fl(2)d*, *nvr3*, *gish*, *Atpalpha*, *CG2991*, *pum*, *Syx1A*, *S6k*, *cv-2*, *MESK2*. In **l**, *P* > 0.9999, =0.2282, 0.0832, 0.0939, 0.5032, 0.455, 0.8416, 0.7415 for *Oda*, *Atpalpha*, *Syx1A*, *gish*, *MESK2*, *cv-2*, *CG2991*, *fl(2)d*. Source data are provided as a Source Data file.

shed light on how the m6A modification contributes to splicing regulation. In all five cases, we analyzed, four (*Dsp1*, *CG8929*, *fl(2)d*, *Aldh-III*) in 5′UTRs and one (*Sxl*) in exon/intron, reduction of m6A modification was correlated with enhanced splicing, arguing that the normal role of these modifications might be to repress splicing events nearby.

Last but probably the most interesting finding from our work is to demonstrate the female-specific m6A modification around *Sxl* exon3. *Sxl* is a textbook paradigm to study alternative splicing and has been intensively investigated for more than thirty years[69]. Sxl protein binds to its own mRNA to control the alternative splicing, but its binding sites are located >200 nucleotides downstream or upstream of the male exon, meaning other regulators should be involved. Recently, the m6A modification pathway was shown to modulate *Sxl* alternative splicing, but the detailed mechanism has not been resolved[33,47,48]. Our MeRIP-seq data revealed that several m6A peaks were deposited only in females on and around *Sxl* exon3, and they were in the vicinity of Sxl-binding sites (Fig. 6e). We further validated this finding by independent m6A-IP-qPCR and showed that these modifications were reduced in *Mettl3* mutant females. This unexpected finding suggests a model that one main function of Sxl may be to recruit the m6A writer complex that methylates nearby m6A sites. The m6A reader Ythdc1 in turn binds to these sites and might interact with the splicing machinery to repress splicing (Fig. 6h). Future experiments, such as interactions between Sxl and Mettl3/Mettl14, interactions between Ythdc1 and general splicing factors, mapping of the exact m6A methylation site in *Sxl* at the single nucleotide level, comparison of transcriptome-wide binding sites of Sxl with m6A modification sites, will be required to firmly prove our model.

## Methods

**Fly strains and genetics.** Flies were grown on standard cornmeal food and experiments were performed at 25 °C. The following stocks were used: *w^{1118}* (used as wild-type, WT), *yw*, *Mettl3^{SK2}*, *Mettl14^{SK1}* [47], *Df(3R)Exel6197* (*Mettl3* deficiency, Bloomington 7676), *ap-Gal4*, *Hakai* shRNA (VDRC330548), *vir shRNA* (HMC03908, Bloomington 55694), *fl(2)d shRNA* (HMC03833, Bloomington 55674), *flacc shRNA* (VDRC 35212GD), *actin-Cas9* (Bloomington 54590)[55], *nanos-Cas9* (Bloomington 78782)[57]. To generate *U6-Hakai-sgRNA*, target sequence "ACGTCCGCGCCCGCGAGCCC" was picked by DRSC Find CRISPRs[57] and cloned into pCFD3 vector[55] (Addgene 49410). The construct was inserted at attP40 site by standard PhiC31-mediated transformation to make transgenic flies in UniHuaii. *U6-Hakai-sgRNA* was crossed with *nanos-Cas9* flies to generate a series of indel mutations, and *Hakai^{SH2}* and *Hakai^{SH4}* were chosen for further analysis. Two additional Hakai shRNAs were made based on the pNP vector by Tsinghua Fly Center[56]. TH14422.S with target sequence "GAGCTCGACAAGGACGGCGAA" was inserted at attP2 site and TH14423.S with target sequence "CGGCCGCATGATACCCTGCAA" was inserted at attP40 site. The three *Hakai* shRNAs show similar knockdown efficiency as examined by Hakai antibody staining. VDRC 330548 was used in Figs. 4g, j, k, 5c–c″, TH14423.S was used in Fig. 4h–h′, 4i, and TH14422.S was used in Fig. 4l. To test adult flight ability, cohorts of ten male flies

were tapped down into a Petri dish, and the number of flies that flew away within 2 min was recorded.

**Immunostaining.** Wing discs from 3rd instar larvae were dissected in PBS and fixed in 4% formaldehyde (Sigma) in PBST (PBS + 0.1% Triton X-100) for 15 min. After blocking in 1% normal donkey serum (Jackson Immuno) in PBST for 1 h, the samples were incubated with the primary antibody in the same solution at 4 °C overnight. After three washes in PBST, samples were incubated with the secondary antibody for 2 h at room temperature, washed in PBST three times, and subsequently mounted in Antifade Mounting Medium (Beyotime). All images were taken on a Zeiss LSM 880 microscope.

The following antibodies were used: mouse anti-Fl(2)d (1:10) (9G2, DSHB), rat anti-Ecad (1:5) (DCAD2, DSHB), mouse anti-Sxl (1:10) (M18, DSHB), rabbit anti-Flacc (1:200)[36], rabbit anti-Hakai (1:200), rabbit anti-Vir (1:200), rabbit anti-Mettl3 (1:200), rabbit anti-Mettl14 (1:200); Alexa 488-conjugated anti-mouse and anti-rabbit IgG (1:1000) (Thermo Fisher, A21202, A21206), Cy3-conjugated anti-mouse, and anti-rabbit IgG (1:400) (Jackson Immuno, 715-165-150, 711-165-152). Hakai antibody was generated in rabbits against a recombinant protein containing amino acids 1–120. Vir antibody was generated in rabbits against a recombinant protein containing amino acids 750–1005. Mettl3 antibody was generated in rabbits against a recombinant protein containing amino acids 110–300. Mettl14 antibody was generated in rabbits against a recombinant protein containing amino acids 1–205. All four antibodies were generated and affinity-purified at ABclonal.

**Molecular cloning and co-immunoprecipitation.** To generate the GFP-, mRFP-, or HA-tagged plasmids, full-length cDNAs for Hakai (the long and the short isoform amplified from a cDNA library), Flacc (GH14795), Nito (GH11110), Fl(2)d (LD21616), METTL3 (AT20169), and METTL14 (LD06016) were cloned into the pENTR vector (Invitrogen), and transferred into the *Drosophila* Gateway vector pAGW, pARW, and pAHW. GFP was cloned into pAWM as a control.

*Drosophila* S2 cells were maintained in Schneider's medium (Gibco) supplemented with 10% FBS (Gibco) at 25 °C. In all, 4 µg of total DNA was transfected into S2 cells in a 100-mm dish with Effectene (QIAGEN). After 48 h, cells were lysed in IP lysis buffer (Beyotime, 50 mM Tris (pH 7.4), 150 mM NaCl, 1% NP-40, 1× protease inhibitor) on ice for 30 min, and cleared at 20,000×g for 10 min at 4 °C. Supernatants were incubated with anti-GFP nanobody agarose beads (Allele Biotechnology) for 2 h at 4 °C. The beads were washed three to four times with 1 ml lysis buffer and resuspended in 2× SDS sample buffer. Eluted proteins were detected by western blotting using rabbit anti-GFP (1:1000, A6455, Molecular Probes), rat anti-HA (1:1000, 3F10, Roche) or mouse anti-Fl(2)d (1:100, 9G2, DSHB) primary antibodies, and HRP-conjugated secondary antibodies (1:3000, Santa Cruz Biotech). Uncropped western blots can be found in the Source Data file. For co-localization, S2 cells were grown on Lab-Tek chamber slides and imaged in live conditions 2 days after transfection. HeLa cells were cultured at 37 °C with 5% CO2 in Dulbecco's modified eagle's medium (TransGen) supplemented with 10% FBS (Gibco).

**Analyzing m6A levels by LC-MS.** Total RNAs were extracted from adult flies or cultured cells using TRIzol (Invitrogen), and then subjected to two rounds of poly (A) selection using the GenElute mRNA Miniprep kit (Sigma). Before LC-MS analysis, all RNA samples were hydrolyzed enzymatically to ribonucleosides. Briefly, 600 ng RNA of each sample was digested by nuclease P1 (Sigma) and snake venom phosphodiesterase (Sigma) at 37 °C for 2 h, and followed by digestion with fast alkaline phosphatase (Thermo Fisher) for another 1 h. Then, the digested samples were used for the following LC-MS analysis.

Quantitative LC-MS analyses of m6A and adenosine were achieved using a Waters UPLC coupled to Thermo Q Exactive mass spectrometer in positive ion mode using dynamic multiple reaction monitoring. The ribonucleosides in the hydrolyzed RNA samples were resolved on an Acquity UPLC HSS T3 column (1.8

μm particle size, 100 Å pore size, 2.1 × 100 mm, 30 °C) at 300 μl min$^{-1}$ using a solvent system of 0.1% formic acid in $H_2O$ (A) and acetonitrile (B). The elution profile was 2% B for 2 min, 2–11% B over 4 min, then to 11–80% B over 4 min, followed by a column washing at 80% B and column equilibration. The quantification of a ribonucleoside can be achieved using $m/z$ of the parent ribonucleoside ion and m/z of the deglycosylated ion product. Nucleosides were quantified based on the transition of the parent ribonucleoside to the deglycosylated base ion: $m/z$ 268.1–136.1 for A, $m/z$ 282.1–150.1 for m$^6$A, $m/z$ 282.1–150.1 for m$^1$A, $m/z$ 244.1–112 for C, $m/z$ 258.1–126 for m$^5$C, and $m/z$ 286.1–154 for ac$^4$C. Absolute quantities of each ribonucleoside were determined using an external calibration curve prepared with A (Sigma), m$^6$A (Selleck), m$^1$A (J&K), C (Sigma), m$^5$C (Sigma), and ac$^4$C (J&K) standards.

**MeRIP-seq**. MeRIP-seq was performed following previous protocols[70,71]. The total RNA was isolated from *yw* (male and female), *Mettl3$^{SK2}$/Df* (male), *Mettl14$^{SK1}$* (male), and *Hakai$^{SH4}$* (male) adult fly about 1–2 days old using TRIzol (Invitrogen). The RNA amount of each sample was quantified using NanoDrop ND-1000 and the RNA integrity was assessed by Bioanalyzer 2100 (Agilent) with RIN number >7.0, and confirmed by electrophoresis with denaturing agarose gel. Poly (A) RNA was purified from 50 μg of the total RNA using Dynabeads Oligo (dT)25 (Thermo Fisher) by two rounds of purification and then was fragmented into small pieces using Magnesium RNA Fragmentation Module (NEB) under 86 °C for 7 min. The cleaved RNA fragments were incubated for 2 h at 4 °C with Dynabeads (Dynabeads Antibody Coupling Kit, Thermo Fisher) coupled with m$^6$A-specific antibody (202003, Synaptic Systems) in IP buffer (50 mM Tris-HCl, 750 mM NaCl, and 0.5% Igepal CA-630). Then the IP RNA fragments and untreated input control fragments are converted to the final cDNA library in accordance with a strand-specific library preparation by the dUTP method. The average insert size for the final cDNA library was 200 ± 50 bp. At last, we performed the 2 × 150 bp paired-end sequencing (PE150) on an Illumina Novaseq 6000 (LC-Bio Technology) following the vendor's recommended protocol.

**Bioinformatic analysis**. fastp (v0.19.4) software was used to remove the reads that contained adaptor contamination, low-quality bases, and undetermined bases with default parameters. Then sequence quality of IP and Input samples were also verified using fastp. We used HISAT2 (v2.0.4)[72] to map reads to the genome of *Drosophila_melanogaster* (Version: v96) with default parameters. Mapped reads of IP and input libraries were provided for R package exomePeak (v1.9.1)[73], which identifies m$^6$A peaks with bed or bigwig format that can be adapted for visualization on the IGV software. HOMER (v4.1)[74] was used for de novo and known motif finding followed by localization of the motif with respect to peak summit. Called peaks were annotated by the intersection with gene architecture using R package ChIPseeker (v1.18.0)[75]. A common peak was picked if the peaks from different groups overlap in the genome >50% of the smallest of the peaks. StringTie (v1.3.4)[76] was used to perform expression levels for all mRNAs from input libraries by calculating FPKM. The differentially expressed mRNAs were selected with log2 (fold change) ≥1 or log2 (fold change) ≤ −1 and P value <0.05 by R package edgeR (v4.1)[77]. rMATS (v4.0.1) was used for differential splicing analysis with a filter for FDR < 0.05 and IncLevelDifference ≥0.2 or ≤ −0.2[78].

**RT-PCR**. Total RNAs were extracted using TRIzol (Invitrogen) and cDNAs were generated from 0.5 μg of RNA using Hifair II 1st Strand cDNA Synthesis Kit with gDNA digester plus (Yeasen). 2xHieff PCR Master Mix (Yeasen) was used for regular PCR, and Hieff qPCR SYBR Green Master Mix (Yeasen) was used for qPCR. All primers used are listed in Supplementary Data 9. *Sxl* primers used in Fig. 3b are described in ref. [59] and *Dsp1, CG8929, Aldh-III* and *fl(2)d* primers used in Fig. 3c are described in refs. [33,34]. Statistical analysis was performed using GraphPad Prism (v9.0.0).

**m$^6$A-IP-qPCR**. m$^6$A-IP-qPCR was performed following a protocol with minor modifications[70]. Total RNAs were extracted using TRIzol (Invitrogen) and purified through GenElute mRNA Miniprep kit (Sigma). In total, 5 μg of purified mRNAs were fragmented into ~200 300 nt fragments by incubation in RNA fragmentation reagent (Thermo Fisher) at 94 °C for 30 s and then stopped with stop solution. Ten percent of the fragmented mRNAs were kept as input. The remaining mRNAs were incubated with 6.25 μg m$^6$A-specific antibody (202003, Synaptic Systems) or rabbit IgG (Beyotime) in IP buffer (50 mM Tris-HCl (pH 7.4), 750 mM NaCl, 0.5% Igepal CA-630, 0.4 U/μl RNasin (Promega)) for 2 h at 4 °C. Then the mixture was incubated with 25 μl Dynabeads Protein A (Thermo Fisher) for another 2 h at 4 °C. After extensive washing, the bound mRNAs were eluted with 6.7 mM N6-methyladenosine (Sigma) in IP buffer and then recovered with ethanol precipitation. The immunoprecipitated mRNAs and input mRNAs were processed as in RT-PCR.

**mRNA decay assay**. To assay for mRNA stability, *yw* or *Mettl3* mutant L3 larval imaginal discs were dissected and treated with 20 μg/ml Actinomycin D (Sigma) in Schneider's medium (Gibco) for 4 h at 25 °C. Total RNA was extracted at $T = 0$ h and $T = 4$ h and processed as in RT-PCR.

**Reporting summary**. Further information on research design is available in the Nature Research Reporting Summary linked to this article.

## Data availability
High-throughput m$^6$A-seq and RNA-Seq data are deposited into the Gene Expression Omnibus with accession number GSE155662. The data supporting the findings of this study are available from the corresponding authors upon reasonable request. Source data are provided with this paper.

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

## Acknowledgements

We thank Eric Lai, Matthias Soller, Bloomington Drosophila Stock Center, TsingHua Fly Center, and Vienna Drosophila Resource Center for fly stocks; Developmental Studies Hybridoma Bank for antibodies; Xiaoyan Xu, Wenli Hu, and Shuining Yin from the mass-spec and imaging core facility of Institute of Plant Physiology and Ecology for technical support; Siyuan Liang for help in bioinformatic analysis; Ning Zheng for critical reading of the paper; the referees for suggestions regarding methylation patterns over the *Sxl* gene. This work is supported by the National Key R&D Program of China (2018YFA0800100) and National Natural Science Foundation of China (91857114, 31970786, 31771586) to D.Y.

## Author contributions

Y.W. and D.Y. designed the research; Y.W., L.Z., H.R., L.M., J.G., D.M., L.L., and D.Y. performed the experimental studies; Y.W., Z.L., and D.Y. carried out bioinformatic analysis; Y.W. and D.Y. analyzed the data; and D.Y. wrote the paper.

## Competing interests

The authors declare no competing interests.
