## [Peer Review File · Nature Communications]

REVIEWER COMMENTS

Reviewer #1 (Remarks to the Author):

The m6A writer complex consists of the heterodimer Mettl3/Mettl14, of which Mettl3 is the active enzyme. Additional proteins are part of this large complex. This manuscript describes the study of one of the additional components called Hakai. The model used for the studies is *Drosophila*.

They show that fly Hakai is part of the m6A writer complex. They created loss-of-function mutants of Hakai in *Drosophila* using CRISPR/Cas9. They demonstrate that the male-specific splicing pattern of the Sex lethal gene is now also seen in the Hakai mutant female flies. This is similar to that previously shown for Mettl3 and Mettl14 KO flies. They quantify the ability to fly, and similar to Mettl3 mutant, the Hakai mutants are flightless. They map m6A in the wildtype flies and various other mutants to show that there is a slight reduction in the distribution of m6A across various transcripts. They demonstrate that lack of Hakai leads to reduction in levels of some of the other components in the writer complex. Deep sequencing of mutant flies to identify defects in transcript regulation is also shown.

Most of these results mirror the conclusions reached by the authors in their previous study on another component (Xio) of the writer complex (Guo et al., 2018, PNAS).

This will be useful for the RNA modification community.

Major:

1. Figure-1: There is almost no interaction (panel F, G, I, J) between GFP-Hakai and other m6A writer machinery, except for Fl(2)d/WTAP (panel H). Is the GFP-tag causing steric hindrance for this interaction? Why not check for interaction of endogenous proteins using an HA (small tag) immunoprecipitation? It could be added to this panel. The authors have access to antibodies for Xio from their previous work, and also used WTAP antibody in the previous work.
2. Figure-4A: Why is there is no loss of m6A in the Mettl3 mutant (which is the writer) at the 3' end? Are the Mettl3 mutant flies really complete nulls? Unlike what is mentioned in the paper, there is a clear distribution of the m6A profile towards the 3' UTR, that is consistent with what is known in other systems, but why is the profile almost same in the WT and Mettl3 mutant at the 3' end? The input and IP values should be in the same track (as lines) to appreciate the enrichment. I think this representation style could be improved. It is a bit confusing to appreciate the real enrichment.
3. Figure4H. The authors could add the methylation patterns over the *sxl* gene.
4. Figure 5: RNAseq of whole flies is not informative. Fly head or ovary could be more focused and informative analysis.
5. Figure 6: Is there any change in sub-cellular (nuclear vs cytosolic) distribution of other components in the absence of Hakai? A zoom-in of single cells would be useful.

Minor

1. Figure-3. Indicate the lane with the mis-splicing in females with a box or a small arrowhead, to draw attention of the reader to the striking effect in the mutant.
2. Some comment about E3 ligase activity of Hakai would be useful. Is the domain active? An alignment with active ligases could be shown.

Reviewer #2 (Remarks to the Author):

The manuscript, "Role of Hakai in m6A modification pathway in *Drosophila*" by Yanhua Wang et al. addresses three major questions, Is Hakai involved in m6A modification as part of the writer complex, what are the in vivo targets of Hakai and other components of the pathway, and does Hakai play a role in the destruction of E-cadherins, as proposed in a number of earlier works? The paper provides novel, interesting and important results that should be on considerable interest to those interested in RNA modification. It makes a strong case that Hakai functions in at least some writer complexes, that

it's role has in vivo relevance, and that the earlier proposed role for the gene product, as a regulator of E-cadherin levels, is almost certainly incorrect. The paper appears to add new information and levels of complexity to how m6A modification is controlled in the cell through me-RIP experiments, and offers additional evidence that m6A modifications affect mRNAs involved in many different cellular functions.

The most compelling data in the paper address the question of whether Hakai functions in the writer complex. The authors use, or cite, a variety of protein interaction assays, including mass spec, affinity purification, and co-IP assays. The data suggest that Hakai interacts more strongly with the Fl(2)d component of the writer complex than with other components suggesting perhaps that it interacts directly with Fl(2)d or that there may be more diversity in the components of the writer complex than usually appreciated. (The authors don't really address the possibility that multiple kinds of writer complexes exist, but I would encourage them to consider doing so in a revised manuscript.) The authors also perform colocalization experiments that mark all the members of the complex as being generally distributed nuclear proteins. The broad nuclear distribution limits the power of colocalization as pretty much any broadly distributed protein would show the same pattern, but it is consistent with expectations.

While the protein interaction data are strongly suggestive, the in vivo analysis with hakai mutants makes a compelling case that the gene product functions in m6A modification. Critically, in hakai mutant adults the m6A modification is reduced about half. Even more important are the data showing that Sxl splicing is altered such that the female-specific exon 3 is skipped some of the time in hakai mutant females. So far as is known, only m6A modification defects, or Sxl protein defects result in this phenotype. Hakai mutants show two additional, albeit less specific and well defined, phenotypes characteristic of m6A modification defects, held out wings, and flightlessness. Taken together it is difficult to imagine an alternative explanation for the findings.

The convincing genetic demonstration that hakai defects do not affect E-cadherin levels in vivo is also an important contribution. Earlier experiments had shown that the proteins interact, and the notion that Hakai regulates E-cadherin has apparently become dogma in the field despite the lack of convincing supportive evidence. The genetic analysis done here adds to earlier work in *Drosophila* that failed to find evidence for such an interaction. One hopes with this demonstration that notion will finally be dispelled.

The other major contributions of the paper involve analysis of mRNA modification. This is an area much farther from my expertise and I cannot productively comment on the methods or statistical analysis. The mutant analysis that suggests 5' end modifications are the key contributions of the writer complex is intriguing. I can't see that the GO and KEGG term analysis adds anything more to the paper than would a simple statement that the data suggest m6A modified messages are involved in many cellular processes. If you are looking to shorten the paper or simply the figures deleting the GO and KEGG analysis would be a good place to start.

The one thing I think is missing from the m6A RNA analyses is a specific examination of Sxl mRNA. It's the one message where an effect was specifically measured and visible but the expectations for don't necessarily fit with idea of 5' modifications drawn from the bulk analysis. It would be useful to analyze how the different mutants affect Sxl mRNA m6A modification. Do they all act reduce methylation near the alternative exon or do some modify the 5' end? Could such difference explain their differing effects on Sxl splicing or might they reveal unexpected mechanistic details as to how the splice is regulated? Large scale bulk data have value but very often more important lessons are learned from the analysis of a few critical targets.

I have only one serious criticism of the paper. It is poorly written and full of grammatical errors. Some of the problems are minor and merely annoying. Others obscure the authors' meaning and some end up inflating the authors' claims beyond what is evidenced. The abstract illustrates some of the

problems. Minor typos lines: 27 typicAL, line 33 codonS, line 34 high-confidENCE. Unclear meaning, lines 28,29, 30, where I think they want to state that hakai mutants have defects in common with Metll14 mutants. Lines 36, 37 where I think they want to say Hakai is needed to maintain proper levels of several other components of the writer complex.

Over claiming, which I believe stems from poor English rather than deliberate intent, is evident in the Results, "Hakai is required for m6A methylation in Drosophila." Presumably what the authors mean is that Hakai is involved in m6A methylation as methylation is reduced but not eliminated in hakai mutants. "Hakai controls Sxl alternative splicing and adult fly phenotypes." Once again, the evidence suggests loss of hakai effects these processes but that is not what is generally meant by the word control.

Reviewer #3 (Remarks to the Author):

The m6A writer is a large complex. Hakai was shown in some protein interaction studies as a component of the m6A writer. In this manuscript, Wang and colleagues studied the role of Hakai in fly. They showed typical m6A deficiency phenotype of Hakai mutant, reduced m6A levels, as well as other biochemical evidence supporting Hakai to be a part of m6A writer. Interestingly, they found Hakai may direct m6A distribution pattern on mRNAs. The manuscript is well-organized and the key idea is expressed clearly. I have some questions/suggestions as below:

1. In figure 2, the m6A levels are much lower than previously reported. Please explain why.
2. To show splicing change, high-throughput sequencing analysis is highly recommended in addition to PCR. Whether other genes besides Sxl also worth investigating.
3. They found m6A enriches not only on 3' end, but also on 5' end. This is interesting but needs careful interpretation. High-stringent measurement using independent methods will strengthen the observation.
4. The 5' peaks were significantly reduced in m6A deficient samples while more 3' peaks were preserved. What does it mean? GO/KEGG analysis cannot provide much understanding. The authors also stated in their manuscript: it is not surprising that this modification is involved in numerous biological processes.
5. The RNA-seq analysis of m6A mutant samples was quite superficial. To study the potential function of m6A mediated RNA decay, the RNA life time tracking experiment is needed.
6. The mechanism of Hakai in m6A methylation still needs further studies. This part might be the most important contribution of this manuscript. However, the present data is far from sufficient. Not only the phenotype, but also the molecular mechanism behind. I would expect deep investigation of Hakai as a potential key component of m6A writer.
7. The appearance of figure 6 is quite rough.

Reviewer #1 (Remarks to the Author):

The m6A writer complex consists of the heterodimer Mettl3/Mettl14, of which Mettl3 is the active enzyme. Additional proteins are part of this large complex. This manuscript describes the study of one of the additional components called Hakai. The model used for the studies is Drosophila.

They show that fly Hakai is part of the m6A writer complex. They created loss-of-function mutants of Hakai in Drosophila using CRISPR/Cas9. They demonstrate that the male-specific splicing pattern of the Sex lethal gene is now also seen in the Hakai mutant female flies. This is similar to that previously shown for Mettl3 and Mettl14 KO flies. They quantify the ability to fly, and similar to Mettl3 mutant, the Hakai mutants are flightless. They map m6A in the wildtype flies and various other mutants to show that there is a slight reduction in the distribution of m6A across various transcripts. They demonstrate that lack of Hakai leads to reduction in levels of some of the other components in the writer complex. Deep sequencing of mutant flies to identify defects in transcript regulation is also shown.

Most of these results mirror the conclusions reached by the authors in their previous study on another component (Xio) of the writer complex (Guo et al., 2018, PNAS).

This will be useful for the RNA modification community.

We thank the reviewer for the positive comments and insightful suggestions.

Major:

1. Figure-1: There is almost no interaction (panel F, G, I, J) between GFP-Hakai and other m6A writer machinery, except for Fl(2)d/WTAP (panel H). Is the GFP-tag causing steric hindrance for this interaction? Why not check for interaction of endogenous proteins using an HA (small tag) immunoprecipitation? It could be added to this panel. The authors have access to antibodies for Xio from their previous work, and also used WTAP antibody in the previous work.

We thank the reviewer for the constructive suggestions. Indeed, interactions in our initial version of Fig 1. seemed weak, so we repeated these experiments with more cells (100mm dish versus 60mm dish before). This time we kept the IPs with similar level of exposure, so it can be appreciated that pull down was strongest for Fl(2)d, and weakest for Nito (Fig. 1F).

As suggested, we have constructed HA-Hakai and used HA-agarose (Sigma) for our immunoprecipitation. However, we found that it is less effective compared to GFP-nanobody trap that provides strong pulldown. Thus, we think it is unlikely the GFP-tag causes steric hindrance for the interaction. In addition, HA pulldown was interfered with IgG bands that are close to Fl(2)d in the protein gel.

We also tested interactions of endogenous proteins since we generated more antibodies for m⁶A writers in rabbits. Unfortunately, all polyclonal antibodies gave many non-specific bands in western blot, even if they are quite good for immunostaining. The only antibody that worked well in western blot is the monoclonal FI(2)d antibody. So we further examined the ability of GFP-Hakai to pull down endogenous FI(2)d protein. Hakai has a long and a short protein isoform, both of which contain a RING finger domain and a C2H2 domain. We also included the short isoform in our assay and found that both GFP-Hakai (long isoform) and GFP-Hakai-S (short isoform) can robustly pull down FI(2)d to a similar extent. These data have been added as Fig. 1G.

2. Figure-4A: Why is there is no loss of m⁶A in the Mettl3 mutant (which is the writer) at the 3' end? Are the Mettl3 mutant flies really complete nulls? Unlike what is mentioned in the paper, there is a clear distribution of the m⁶A profile towards the 3' UTR, that is consistent with what is known in other systems, but why is the profile almost same in the WT and Mettl3 mutant at the 3' end? The input and IP values should be in the same track (as lines) to appreciate the enrichment. I think this representation style could be improved. It is a bit confusing to appreciate the real enrichment.

We are sorry for the confusion. In the original peak density plots, the distribution for each sample has been normalized so that the areas under the line remain the same. So, such density plots cannot be used to compare methylation levels between samples, and they can only demonstrate the relative distribution of m⁶A peaks along the transcript. In the new version, we have separated these density plots into individual panel to avoid confusion (Fig. 6A, 7A-C).

We used previously reported alleles *Mettl3*^{SK2} and *Mettl14*^{SK1} from Eric Lai lab. Both of them are early frame-shift mutations and were characterized as strong alleles (Kan, et al, Nat. Commun., 2017). The phenotypes of *Mettl3*^{SK2} adult flies are similar to those of another null allele of *Mettl3* obtained from Matthias Soller lab. Furthermore, we tested these alleles by antibody staining and found complete loss of Mettl3 and Mettl14 antibody staining in these mutant discs (Supplemental Fig. 5).

We have mentioned in the paper that most m⁶A peaks were enriched in the 3'UTR and close to the stop codon, and to a lesser extent enriched in the 5' UTR and around the start codon (Fig. 6A, B), consistent with other systems. Since most m⁶A peaks reduced in m⁶A mutants are located in 5' UTRs (compare 7A with 6A), distribution in 3' UTRs appears higher as this is a normalized plot.

We have included the distribution of mapped reads for IP and input samples in the same scale (Supplemental Fig. 6), and it can be seen that reads were enriched

around TSS and TES in IP samples compared to inputs, validating our IP experiments.

To compare methylation levels, cumulative distribution plot demonstrated that there are less fold enrichment (m^6A -IP/input) in m^6A mutants than *yw*, implying less methylation levels (Fig. 7E). Indeed, when we filtered m^6A peaks with higher stringency (fold enrichment \geq 5), the number of m^6A peaks dropped more than 40% in *Mettl3* mutant compared with *yw* (Fig. 7F). In addition, many more m^6A peaks were reduced (*Mettl3*, 2745; *Mettl14*, 2615; *Hakai*, 2036; $p < 0.05$ and fold change ≤ 0.5) than increased (*Mettl3*, 330; *Mettl14*, 260; *Hakai*, 278, $p < 0.05$ and fold change ≥ 2) in these mutants, validating their role as m^6A writers.

3. Figure 4H. The authors could add the methylation patterns over the *Sxl* gene.

We really appreciate the reviewer's comments on methylation patterns over *Sxl*. We performed MeRIP-seq for five samples simultaneously: *yw* female, *yw* male, *Mettl3* male, *Mettl14* male, and *Hakai* male. In the initial submission of our manuscript, we only included four data sets for male flies, since we did not pay much attention to the *Sxl* locus. Because previously the methylation sites for *Sxl* were mapped in introns and our MeRIP-seq was done in mRNAs after one round of polyA selection, I did not expect to see much methylation on this region.

However, inspired by the reviewers' comments, we focused on the *Sxl* locus, especially around the male-specific exon3, for potential m^6A sites. Strikingly, we found 3-4 m^6A peaks in and around exon3 that only exist in female flies, but not males. We further validated the MeRIP-seq results with independent m^6A -IP-qPCR and these m^6A modifications were strongly reduced in *Mettl3* females. These methylation patterns have been added to the manuscript and actually been expanded to the new Figure 6. Our results demonstrated that the m^6A modifications are deposited in a sex-specific manner, which has not been shown in *Drosophila* or any other species before. Based on this finding, we developed a model to explain how the m^6A modifications cooperate with *Sxl* protein to regulate its mRNA splicing.

4. Figure 5: RNAseq of whole flies is not informative. Fly head or ovary could be more focused and informative analysis.

We agree with the reviewer that RNA-seq of whole flies is not particularly informative, and I also had experience that large amount of RNAs from ovaries obscured the gene expression from other tissues. However, our RNA-seq was part of the MeRIP-seq experiment and was used as input in this analysis, thus RNA-seq from other tissues cannot be combined with MeRIP-seq data. We used male adult flies for comparison that are less influenced by the large amount of RNAs from ovaries. In addition, RNA-seq were done in WT and *Mettl3* mutant fly heads before (Hausmann, et al, Nature, 2016). We think that RNA-seq from whole flies can also provide some new

information, for example, we found strong upregulation of immune response genes and antimicrobial peptides in our mutants, which was not reported from previous fly head analysis.

5. Figure 6: Is there any change in sub-cellular (nuclear vs cytosolic) distribution of other components in the absence of Hakai? A zoom-in of single cells would be useful.

To further study the role of Hakai in the writer complex, we generated antibodies against Mettl3, Mettl14 and Vir, constituting a full toolkit for all seven *Drosophila* m⁶A writers. Knocking down *Hakai* led to no effect on Mettl3, Mettl14 and Nito protein levels, but strong reduction of Fl(2)d, Vir and Flacc levels (Fig. 4G-L). For comparison, we knocked down *Flacc* that was proposed to anchor the writer complex in the nucleus, and observed more diffusive and less nuclear staining of Fl(2)d (Fig. 4D-E). Areas from these images were magnified to show individual cells. These data suggest that the main effect in the absence of Hakai is reduced protein levels rather than change in sub-cellular distribution for other components.

Since the roles of several other m⁶A writer components are not fully understood, we extended our immunostaining assays to other genes. Interestingly, knocking down *vir* by RNAi resulted in no effect on Mettl3, Mettl14 and Nito protein levels, but strong reduction of Fl(2)d, Hakai and Flacc levels. Similarly, depletion of *fl(2)d* by RNAi did not change the protein levels of Mettl3, Mettl14 and Nito, but strongly reduced Hakai, Vir and Flacc levels. Together, these results suggest that Fl(2)d, Vir, Hakai and Flacc form a stable complex and disruption of either Fl(2)d, Vir or Hakai leads to degradation of the whole complex, while Mettl3, Mettl14 and Nito are not part of this complex. Based on our new data, we proposed a working model for the m⁶A writer complex (Fig. 4Y).

Minor

1. Figure-3. Indicate the lane with the mis-splicing in females with a box or a small arrowhead, to draw attention of the reader to the striking effect in the mutant.

We have added arrowheads to the lanes with *Sxl* mis-splicing in females. Thanks for the suggestion.

2. Some comment about E3 ligase activity of Hakai would be useful. Is the domain active? An alignment with active ligases could be shown.

We have added some comments about E3 ligase activity of Hakai in the discussion. "Hakai is a potential E3 ubiquitin ligase with an intact C3HC4 RING domain and a C2H2 domain. Its absence led to the degradation, rather than accumulation of other m⁶A writer subunits, indicating that it may not act as a E3 ubiquitin ligase in this complex. Whether Hakai has other substrates for its E3 ligase activity needs to be

determined.” We also included an alignment of Hakai with human CBLL1 and CBLL2 showing active domains (Supplementary Fig. 2A).

Reviewer #2 (Remarks to the Author):

The manuscript, “Role of Hakai in m6A modification pathway in Drosophila” by Yanhua Wang et al. addresses three major questions, Is Hakai involved in m6A modification as part of the writer complex, what are the in vivo targets of Hakai and other components of the pathway, and does Hakai play a role in the destruction of E-cadherins, as proposed in a number of earlier works? The paper provides novel, interesting and important results that should be of considerable interest to those interested in RNA modification. It makes a strong case that Hakai functions in at least some writer complexes, that its role has in vivo relevance, and that the earlier proposed role for the gene product, as a regulator of E-cadherin levels, is almost certainly incorrect. The paper appears to add new information and levels of complexity to how m6A modification is controlled in the cell through me-RIP experiments, and offers additional evidence that m6A modifications affect mRNAs involved in many different cellular functions.

We thank the reviewer for the encouraging comments and insightful suggestions.

The most compelling data in the paper address the question of whether Hakai functions in the writer complex. The authors use, or cite, a variety of protein interaction assays, including mass spec, affinity purification, and co-IP assays. The data suggest that Hakai interacts more strongly with the FI(2)d component of the writer complex than with other components suggesting perhaps that it interacts directly with FI(2)d or that there may be more diversity in the components of the writer complex than usually appreciated. (The authors don't really address the possibility that multiple kinds of writer complexes exist, but I would encourage them to consider doing so in a revised manuscript.) The authors also perform colocalization experiments that mark all the members of the complex as being generally distributed nuclear proteins. The broad nuclear distribution limits the power of colocalization as pretty much any broadly distributed protein would show the same pattern, but it is consistent with expectations.

We thank the reviewer for constructive suggestions and have tried to figure out the mechanisms of Hakai in the m⁶A writer complex. Our improved Co-IP experiments with similar level of exposure indicated that the interaction between Hakai and FI(2)d is the strongest, while its interaction with Nito is the weakest. We further show that GFP-Hakai was able to pull down endogenous FI(2)d and its N-terminal domains are likely important for this interaction.

Since the roles of several other m⁶A writer components are not fully understood, we generated antibodies against Mettl3, Mettl14 and Vir, constituting a full toolkit for all seven *Drosophila* m⁶A writers. Our systematic analysis of several m⁶A writer subunits have provided novel insights into the mechanism of this important complex. We found that Fl(2)d, Vir, Hakai and Flacc form a stable complex, and knocking down either of Fl(2)d, Vir or Hakai led to the degradation of the other three components. Mettl3, Mettl14 and Nito were not affected by the disruption of Fl(2)d, Vir or Hakai, suggesting that they have separate functions. Knocking down Flacc resulted in less nuclear staining of Fl(2)d, consistent with a role in nuclear localization of the writer complex. Based on these results, we proposed a new model for the m⁶A methyltransferase complex (Fig. 4Y). Mettl3 and Mettl14 form a stable heterodimer to catalyze the addition of the methyl group to mRNA. Nito/RBM15 contains three RRM domains and binds to positions adjacent to m⁶A sites, thus may provide target specificity for the m⁶A writer complex. Fl(2)-Vir-Hakai-Flacc form a platform to connect different components and may integrate environmental and cellular signals to regulate m⁶A methylation.

While the protein interaction data are strongly suggestive, the in vivo analysis with hakai mutants makes a compelling case that the gene product functions in m6A modification. Critically, in hakai mutant adults the m6A modification is reduced about half. Even more important are the data showing that Sxl splicing is altered such that the female-specific exon 3 is skipped some of the time in hakai mutant females. So far as is known, only m6A modification defects, or Sxl protein defects result in this phenotype. Hakai mutants show two additional, albeit less specific and well defined, phenotypes characteristic of m6A modification defects, held out wings, and flightlessness. Taken together it is difficult to imagine an alternative explanation for the findings.

We thank the reviewer for the positive comments.

The convincing genetic demonstration that hakai defects do not affect E-cadherin levels in vivo is also an important contribution. Earlier experiments had shown that the proteins interact, and the notion that Hakai regulates E-cadherin has apparently become dogma in the field despite the lack of convincing supportive evidence. The genetic analysis done here adds to earlier work in Drosophila that failed to find evidence for such an interaction. One hopes with this demonstration that notion will finally be dispelled.

We thank the reviewer for pinpointing the importance of our finding to clarify the role of Hakai in E-cadherin regulation.

The other major contributions of the paper involve analysis of mRNA modification. This is an area much farther from my expertise and I cannot productively comment on the methods or statistical analysis. The mutant analysis that suggests 5' end

modifications are the key contributions of the writer complex is intriguing. I can't see that the GO and KEGG term analysis adds anything more to the paper than would a simple statement that the data suggest m⁶A modified messages are involved in many cellular processes. If you are looking to shorten the paper or simply the figures deleting the GO and KEGG analysis would be a good place to start.

We thank the reviewer for the appreciation of our finding and agree that the GO and KEGG term analysis of m⁶A containing genes did not provide much new information. Since there are already quite a lot of new data added to the revised manuscript, we have deleted this part from the paper.

The one thing I think is missing from the m⁶A RNA analyses is a specific examination of Sxl mRNA. It's the one message where an effect was specifically measured and visible but the expectations for don't necessarily fit with idea of 5' modifications drawn from the bulk analysis. It would be useful to analyze how the different mutants affect Sxl mRNA m⁶A modification. Do they all act reduce methylation near the alternative exon or do some modify the 5' end? Could such difference explain their differing effects on Sxl splicing or might they reveal unexpected mechanistic details as to how the splice is regulated? Large scale bulk data have value but very often more important lessons are learned from the analysis of a few critical targets.

We really appreciate the reviewer's insightful comments about Sxl mRNA. We performed MeRIP-seq for five samples simultaneously: yw female, yw male, *Mettl3* male, *Mettl14* male, and *Hakai* male. In the initial submission of our manuscript, we only included four data sets for male flies, since we did not pay much attention to the Sxl locus. Because previously the methylation sites for Sxl were mapped in introns and our MeRIP-seq was done in mRNAs after one round of polyA selection, I did not expect to see much methylation on this region.

However, inspired by the reviewers' comments, we focused on the Sxl locus, especially around the male-specific exon3, for potential m⁶A sites. Strikingly, we found 3-4 m⁶A peaks in and around exon3 that only exist in female flies, but not males. We further validated the MeRIP-seq results with independent m⁶A-IP-qPCR and these m⁶A modifications were strongly reduced in *Mettl3* females. Our results demonstrated that the m⁶A modifications are deposited in a sex-specific manner, which has not been shown in *Drosophila* or any other species before.

How can these m⁶A modifications be installed only in females? Sxl binds to polyU sites located in Sxl intron2 and intron3 and interestingly our mapped m⁶A peaks were close to those Sxl binding sites. In addition, it was known that Sxl physically interacts with four m⁶A writer components, Fl(2)d, Vir, Nito and Flacc. Based on our new data, we developed a model to explain how the m⁶A modifications cooperate with Sxl protein to regulate its mRNA splicing. Sxl in females recruits the m⁶A writer complex that in turn methylates m⁶A sites located in exon3 and nearby introns. Since these

sites are quite close to exon/intron junction regions, m⁶A reader Ythdc1 may bind to these sites and interfere with the splicing machinery, forcing the exon3 to be skipped in females. In support of this view, ectopic expression of Ythdc1 in male discs by *ptc-Gal4* resulted in Sxl protein expression at a level comparable to females, arguing the key regulatory role of m⁶A in this process.

As the reviewer foresightedly pointed out, we do find some unexpected results that shed light on the mechanistic details as to how Sxl splicing is regulated.

I have only one serious criticism of the paper. It is poorly written and full of grammatical errors. Some of the problems are minor and merely annoying. Others obscure the authors' meaning and some end up inflating the authors' claims beyond what is evidenced. The abstract illustrates some of the problems. Minor typos lines: 27 typicAL, line 33 codonS, line 34 high-confidENCE. Unclear meaning, lines 28,29, 30, where I think they want to state that hakai mutants have defects in common with Metll14 mutants. Lines 36, 37 where I think they want to say Hakai is needed to maintain proper levels of several other components of the writer complex.

We are sorry about the grammatical errors and have read through our manuscript several times to correct them. I also asked a professional writer to help us with the grammar mistakes. The problems mentioned have been fixed in the abstract and throughout the main text. We thank the reviewer for a close reading and the comments have helped us to improve the quality of our manuscript significantly.

Over claiming, which I believe stems from poor English rather than deliberate intent, is evident in the Results, "Hakai is required for m6A methylation in Drosophila." Presumably what the authors mean is that Hakai is involved in m6A methylation as methylation is reduced but not eliminated in hakai mutants.

We are sorry about the over claiming problem and have tried to correct such statements throughout the manuscript. This sentence has been changed to "Hakai is required to maintain proper levels of m⁶A methylation".

"Hakai controls Sxl alternative splicing and adult fly phenotypes." Once again, the evidence suggests loss of hakai effects these processes but that is not what is generally meant by the word control.

We have changed this sentence to "Sxl alternative splicing and adult fly behaviour were defective in *Hakai* mutant".

Reviewer #3 (Remarks to the Author):

The m6A writer is a large complex. Hakai was shown in some protein interaction studies as a component of the m6A writer. In this manuscript, Wang and colleagues studied the role of Hakai in fly. They showed typical m6A deficiency phenotype of Hakai mutant, reduced m6A levels, as well as other biochemical evidence supporting Hakai to be a part of m6A writer. Interestingly, they found Hakai may direct m6A distribution pattern on mRNAs. The manuscript is well-organized and the key idea is expressed clearly. I have some questions/suggestions as below:

We thank the reviewer for the positive comments and insightful suggestions.

1. In figure 2, the m6A levels are much lower than previously reported. Please explain why.

We thank the reviewer to point out this important question and we were also puzzled about this issue. We found that the m⁶A levels we measured in *Drosophila* (0.01-0.02% of adenosine after two rounds of polyA purification) are one magnitude lower than those in mammals (0.1-0.4% of adenosine). Our result is consistent with a previous study showing m⁶A represents 0.04% of adenosine after one round of polyA selection in *yw* flies (Kan, et al, Nat. Commun., 2017). Usually, the m⁶A level dropped after each round of polyA purification.

Since the m⁶A level was not firmly determined in *Drosophila*, we measured the m⁶A level, as well as the level of several other RNA modifications such as m¹A, m⁵C, ac⁴C, in *w¹¹¹⁸*, *yw* flies, S2 cells, and human HeLa cells. To our surprise, m⁶A level was 5-10 times higher in human cells than those in *Drosophila*, while m¹A, m⁵C and ac⁴C levels were comparable (Fig. 2G). These results imply that the function and mechanism of m⁶A pathway may be quite different between human and fly. Indeed, *Mettl3* or *Mettl14* mutants are embryonic lethal in mouse while they develop into adults in flies. We have added this interesting new finding to our revised manuscript.

2. To show splicing change, high-throughput sequencing analysis is highly recommended in addition to PCR. Whether other genes besides Sxl also worth investigating.

We performed differential splicing analysis for our RNA-Seq data using rMATS tool. In *Mettl3*, *Mettl14* and *Hakai* mutant flies, 445, 364 and 340 genes were differentially spliced (FDR<0.05, IncLevelDifference≥0.2 or ≤-0.2), respectively, and they overlapped substantially with each other. The differential alternative spliced events occurred in all splicing categories and were functionally enriched for mRNA splicing and signaling transduction, etc. We have added these data to Supplementary Fig. 9.

Other than *Sxl*, it was reported that splicing of several other genes, including *Dsp1*, *CG8929*, *Aldh-III*, and *fl(2)d*, depends on the m⁶A pathway. We then analyzed the splicing isoforms for these transcripts by RT-qPCR. In *Hakai^{SH4}* mutants, the splicing patterns for all four genes were affected similarly to those in *Mettl3* or *Mettl14* mutants (Fig. 3C). These results further strengthen our conclusion and it is worth noting that in all four cases, the spliced isoforms were increased while the unspliced forms were reduced in m⁶A pathway mutants.

3. They found m6A enriches not only on 3' end, but also on 5' end. This is interesting but needs careful interpretation. High-stringent measurement using independent methods will strengthen the observation.

We thank the reviewer for this constructive suggestion. We used independent m⁶A-IP-qPCR to validate the MeRIP-seq results. For *Sxl*, substantial enrichment was observed only in female mRNA IPed with m⁶A antibody, but not in female mRNA IPed with control IgG, nor in male mRNA IPed with either m⁶A or IgG antibody. We also used m⁶A-IP-qPCR to validate the reduction of m⁶A peaks in the 5' UTR region for 11 genes and found significant reduction of m⁶A signal in *Mettl3* mutant versus *yw*. Together, these stringent experiments have confirmed our MeRIP-seq and strengthened our observation. We have added these data to Fig. 6F and 7K.

4. The 5' peaks were significantly reduced in m6A deficient samples while more 3' peaks were preserved. What does it mean? GO/KEGG analysis cannot provide much understanding. The authors also stated in their manuscript: it is not surprising that this modification is involved in numerous biological processes.

We think that majority of the peaks in 3' UTRs may be mediated by another methyltransferase or come from non-specific background, anyway they do not depend on the m⁶A writer complex in *Drosophila*.

m⁶A modification in 3' UTRs usually causes mRNA instability and m⁶A in 5' UTRs is linked to translation enhancement. Our results may imply that the major role of m⁶A modification in *Drosophila* is not on mRNA degradation, but possibly on translation upregulation, which can be tested by combining ribosome profiling and functional analysis of single transcript in the future. The splicing experiments raised by the reviewer do shed light on how the m⁶A modification contributes to splicing regulation. In all five cases we analyzed, four (*Dsp1*, *CG8929*, *fl(2)d*, *Aldh-III*) in 5' UTRs and one (*Sxl*) in exon/intron, reduction of m⁶A modification was correlated with enhanced splicing, arguing that the normal role of these modifications might be to repress splicing events nearby.

Both reviewer 2 and reviewer 3 pointed out that GO/KEGG analysis of m⁶A containing genes cannot provide much understanding. Since there are already quite a lot of new data added to the revised manuscript, we have deleted this part from the paper.

5. The RNA-seq analysis of m6A mutant samples was quite superficial. To study the potential function of m6A mediated RNA decay, the RNA life time tracking experiment is needed.

We performed RNA decay assay for validated m⁶A-containing transcripts. Four hours after transcription inhibition by actinomycin D, we did not observe significant difference of mRNA levels for these genes between *Mettl3* and *yw* imaginal discs (Fig. 7L). These findings are consistent with the notion that effective m⁶A modifications are located in 5' UTRs in *Drosophila*, and thus do not mediate mRNA degradation as in mammalian system.

6. The mechanism of Hakai in m6A methylation still needs further studies. This part might be the most important contribution of this manuscript. However, the present data is far from sufficient. Not only the phenotype, but also the molecular mechanism behind. I would expect deep investigation of Hakai as a potential key component of m6A writer.

We agree with the reviewer that this part is one of the most novel contributions of the manuscript. Since the roles of several other m⁶A writer components are not fully understood, we generated antibodies against Mettl3, Mettl14 and Vir, constituting a full toolkit for all seven *Drosophila* m⁶A writers. Our systematic analysis of several m⁶A writer subunits have provided novel insights into the mechanism of this important complex. We found that Fl(2)d, Vir, Hakai and Flacc form a stable complex, and knocking down either of Fl(2)d, Vir or Hakai led to the degradation of the other three components. Mettl3, Mettl14 and Nito were not affected by the disruption of Fl(2)d, Vir or Hakai, suggesting that they have separate functions. Knocking down Flacc resulted in less nuclear staining of Fl(2)d, consistent with a role in nuclear localization of the writer complex. Based on these results, we proposed a new model for the m⁶A methyltransferase complex (Fig. 4Y). Mettl3 and Mettl14 form a stable heterodimer to catalyze the addition of the methyl group to mRNA. Nito/RBM15 contains three RRM domains and binds to positions adjacent to m⁶A sites, thus may provide target specificity for the m⁶A writer complex. Fl(2)-Vir-Hakai-Flacc form a platform to connect different components and may integrate environmental and cellular signals to regulate m⁶A methylation.

7. The appearance of figure 6 is quite rough.

We agree with the reviewer that the appearance of Fig. 6 and Fig. 7 after MeRIP-Seq data are not very logical. Thus, in our new version we moved original Fig. 6 and Fig. 7 ahead as new Fig. 4 and Fig. 5. In this way, the first five figures are more focused on Hakai, while the last two figures are more concerned about m⁶A pathway in general.

REVIEWERS' COMMENTS

Reviewer #1 (Remarks to the Author):

I congratulate the authors for the beautiful m6A-IP dataset that demonstrates sex-specific m6A marks in females on the Sxl locus. This is a very important dataset for the entire field.

Other additional experiments and improvements make this an excellent manuscript.

Reviewer #2 (Remarks to the Author):

The revised manuscript is an improvement over the original that satisfactorily addresses my concerns (and I think, those of the other reviewers.). The new version contains an important additional finding that has great potential significance for the RNA methylation field as it offers important new information about the mechanism of Sxl pre-mRNA splicing. What the authors report is that methylation of Sxl exon 3 and its surrounds is female-specific even though past results, and most people's expectations fit with equal methylation in both sexes. I find this result exciting and very sensible as it suggests that Sxl protein itself may be directing the methylation of its pre-mRNA rather than responding to the methylation. This opens up many questions about mechanism of Sxl splicing and offers the possibility that it will drive experiments that dramatically revise this textbook mechanism of splicing regulation.

My only real concern with the new version is with the ythdc1 over-expression experiment that the authors argue supports their Sxl splicing model. I find the results of this experiment problematic and do not think it strengthens the author's proposed model. Rather, I think that, as presented, it actually weakens the model as the result is not clearly and simply predicted by the model. Specifically, the model posits that ythdc1 protein binds to methylated residues in, and surrounding, the Sxl male exon 3, and interferes with the splicing machinery. There is nothing wrong with the model in the abstract; however, I think the simplest prediction of the model would be that over-expressed ythdc1 would have no effect on Sxl splicing in males because the key residues in the male Sxl mRNA are not methylated. Instead, the authors observed that ectopic ythdc1 protein shifts splicing toward the female mode in males.

What I recommend is that the ythdc1 experiment be dropped from the paper. As is, it neither clearly supports the author's Sxl splicing model or contradicts it.

Reviewer #3 (Remarks to the Author):

In this revised version, the author generated antibodies for m6A writer components in fly. It will be a big benefit for the following studies. Using their tool and relevant experiments, they confirmed the writer composition the mutual interaction. This result is solid and conclusive. The author also clarified the m6A level in fly using LC-MS and found that it is one magnitude lower than that in mammal. To avoid ncRNA contamination, they performed two-round polyA purification. I appreciate their efforts, also I strongly recommend further investigation in their future work. A previous study showed higher m6A level in fly. It may reflect more ncRNA contamination, as the author suggested in the manuscript; however, it could also be because of sample variance or dynamics. Another thing worth testing is to enhance the polyA purification, that after 3 or 4 rounds selection, would the m6A level further decrease? If so, the accurate m6A level and its function need reassessment in fly. In sum, most of my concerns have been well addressed. I recommend publication of this manuscript in Nature Communications.

We thank the three reviewers for spending the time and effort in reviewing our manuscript, and their insightful suggestions have greatly improved the quality of our work. Below is our point-to-point response to the reviewers' comments (copied in *blue italics*).

Reviewer #1 (Remarks to the Author):

I congratulate the authors for the beautiful m6A-IP dataset that demonstrates sex-specific m6A marks in females on the Sxl locus. This is a very important dataset for the entire field.

Other additional experiments and improvements make this an excellent manuscript.

We thank the reviewer for the support of our work.

Reviewer #2 (Remarks to the Author):

The revised manuscript is an improvement over the original that satisfactorily addresses my concerns (and I think, those of the other reviewers.). The new version contains an important additional finding that has great potential significance for the RNA methylation field as it offers important new information about the mechanism of Sxl pre-mRNA splicing. What the authors report is that methylation of Sxl exon 3 and its surrounds is female-specific even though past results, and most people's expectations fit with equal methylation in both sexes. I find this result exciting and very sensible as it suggests that Sxl protein itself may be directing the methylation of its pre-mRNA rather than responding to the methylation. This opens up many questions about mechanism of Sxl splicing and offers the possibility that it will drive experiments that dramatically revise this textbook mechanism of splicing regulation.

My only real concern with the new version is with the ythdc1 over-expression experiment that the authors argue supports their Sxl splicing model. I find the results of this experiment problematic and do not think it strengthens the author's proposed model. Rather, I think that, as presented, it actually weakens the model as the result is not clearly and simply predicted by the model. Specifically, the model posits that ythdc1 protein binds to methylated residues in, and surrounding, the Sxl male exon 3, and interferes with the splicing machinery. There is nothing wrong with the model in the abstract; however, I think the simplest prediction of the model would be that over-expressed ythdc1 would have no effect on Sxl splicing in males because the key residues in the male Sxl mRNA are not methylated. Instead, the authors observed that ectopic ythdc1 protein shifts splicing toward the female mode in males. What I recommend is that the ythdc1 experiment be dropped from the paper. As is, it neither clearly supports the author's Sxl splicing model or contradicts it.

We thank the reviewer for the support of our work. Our initial thoughts were that there are residual levels of m⁶A methylation around Sxl in males (Fig. 6e), and overexpression of Ythdc1 might still be able to bind to these sites and interfere with splicing. Indeed, this experiment neither clearly support our model or contradicts it, we included it in the manuscript partially because of the striking male-to-female transformation effect. Following the reviewer's suggestion, we have removed this experiment in our revised version. We are working on more experiments to further prove our model and will include this result for the future story.

Reviewer #3 (Remarks to the Author):

In this revised version, the author generated antibodies for m6A writer components in fly. It will be a big benefit for the following studies. Using their tool and relevant experiments, they confirmed the writer composition the mutual interaction. This result is solid and conclusive. The author also clarified the m6A level in fly using LC-MS and found that it is one magnitude lower than that in mammal. To avoid ncRNA contamination, they performed two-round polyA purification. I appreciate their efforts, also I strongly recommend further investigation in their future work. A previous study showed higher m6A level in fly. It may reflect more ncRNA contamination, as the author suggested in the manuscript; however, it could also be because of sample variance or dynamics. Another thing worth testing is to enhance the polyA purification, that after 3 or 4 rounds selection, would the m6A level further decrease? If so, the accurate m6A level and its function need reassessment in fly. In sum, most of my concerns have been well addressed. I recommend publication of this manuscript in Nature Communications.

We thank the reviewer for the support of our work.